# Discovering Policies with DOMiNO: Diversity Optimization Maintaining Near Optimality

**Tom Zahavy, Yannick Schroecker, Feryal Behbahani, Kate Baumli,
Sebastian Flennerhag, Shaobo Hou and Satinder Singh**
**DeepMind, London**

## Abstract

In this work we propose a Reinforcement Learning (RL) agent that can discover complex behaviours in a rich environment with a simple reward function. We define diversity in terms of state-action occupancy measures, since policies with different occupancy measures visit different states on average. More importantly, defining diversity in this way allows us to derive an intrinsic reward function for maximizing the diversity directly. Our agent, DOMiNO, stands for Diversity Optimization Maintaining Near Optimally. It is based on maximizing a reward function with two components: the extrinsic reward and the diversity intrinsic reward, which are combined with Lagrange multipliers to balance the quality-diversity trade-off. Any RL algorithm can be used to maximize this reward and no other changes are needed. We demonstrate that given a simple reward functions in various control domains, like height (stand) and forward velocity (walk), DOMiNO discovers diverse and meaningful behaviours. We also perform extensive analysis of our approach, compare it with other multi-objective baselines, demonstrate that we can control both the quality and the diversity of the set via interpretable hyperparameters, and show that the set is robust to perturbations of the environment.

## 1 Introduction

As we make progress in Artificial Intelligence, our agents get to interact with richer and richer environments. This means that we cannot expect such agents to come to fully understand and control all of their environment. Nevertheless, given an environment that is rich enough, we would like to build agents that are able to discover complex behaviours even if they are only provided with a simple reward function. Once a reward is specified, most existing RL algorithms will focus on finding the single best policy for maximizing it. However, when the environment is rich enough, there may be many qualitatively (optimal or near-optimal) different policies for maximising the reward, even if it is simple. Finding such diverse set of policies may help an RL agent to become more robust to changes, to construct a basis of behaviours, and to generalize better to future tasks.

Our focus in this work is on agents that find creative and new ways to maximize the reward, which is closely related to Creative Problem Solving (Osborn, 1953): the mental process of searching for an original and previously unknown solution to a problem. Previous work on this topic has been done in the field of Quality-Diversity (QD), which comprises of two main families of algorithms: MAP-Elites (Mouret & Clune, 2015; Cully et al., 2015) and novelty search with local competition (Lehman & Stanley, 2011). Other work has been done in the RL community and typically involves combining the extrinsic reward with a diversity intrinsic reward (Gregor et al., 2017; Eysenbach et al., 2019). Our work shares a similar objective to these excellent previous work, and proposes a new class of algorithms as we will soon explain. Due to space considerations we further discuss the connections to related work in Appendix A.

Our main contribution is a new framework for maximizing the diversity of RL policies in the space of *state-action occupancy measures*. Intuitively speaking, a state-action occupancy $d_\pi$ measures how often a policy $\pi$ visits each state-action pair. Thus, policies with diverse state-action occupancies induce different trajectories. Moreover, for such a diverse set there exist rewards (down stream tasks) for which some policies are better than others (since the value function is a linear product between the state occupancy and the reward). But most importantly, defining diversity in terms of

state occupancies allows us to use duality tools from convex optimization and propose a discovery algorithm that is based, completely, on intrinsic reward maximization. Concretely, one can come up with different diversity objectives, and use the gradient of these objectives as an intrinsic reward. Building on these results, we show how to use existing RL code bases for diverse policy discovery. The only change needed is to to provide the agent with a diversity objective and its gradient.

To demonstrate the effectiveness of our framework, we propose propose **DOMiNO**, a method for Diversity Optimization that Maintains Nearly Optimal policies. DOMiNO trains a set of policies using a policy-specific, weighted combination of the extrinsic reward and an intrinsic diversity reward. The weights are adapted using Lagrange multipliers to guarantee that each policy is near-optimal. We propose **two novel diversity objectives**: a repulsive pair-wise force that motivates policies to have distinct expected features and a Van Der Waals force, which combines the repulsive force with an attractive one and allows us to specify the degree of diversity in the set. We emphasize that our framework is more general, and we encourage others to propose new diversity objectives and use their gradients as intrinsic rewards.

To demonstrate the effectiveness of DOMiNO, we perform experiments in the DeepMind Control Suite (Tassa et al., 2018) and show that given simple reward functions like height and forward velocity, DOMiNO discovers qualitatively diverse and complex locomotion behaviors (Fig. 1b). We analyze our approach and compare it to other multi-objective strategies for handling the QD trade-off. Lastly, we demonstrate that the discovered set is robust to perturbations of the environment and the morphology of the avatar. We emphasize that the focus of our experiments is on validating and getting confidence in this new and exciting approach; we do not explicitly compare it with other work nor argue that one works better than the other.

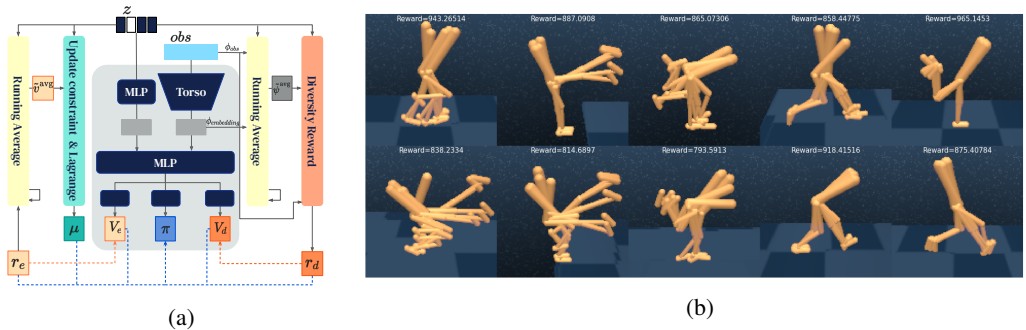

(a)            (b)

Figure 1: **(a) DOMiNO's architecture:** The agent learns a set of diverse high quality policies via a single latent-conditioned actor-critic network with intrinsic and extrinsic value heads. Dashed arrows signify training objectives. **(b) DOMiNO's $\pi$:** Near optimal diverse policies in walker.stand corresponding to standing on both legs, standing on either leg, lifting the other leg forward and backward, spreading the legs and stamping. Not only are the policies different from each other, they also achieve high extrinsic reward in standing (see values on top of each policy).

## 2    PRELIMINARIES AND NOTATION

In this work, we will express objectives in terms of the **state occupancy measure** $d_\pi$. Intuitively speaking, $d_\pi$ measures how often a policy $\pi$ visits each state-action pair. As we will soon see, the classic RL objective of reward maximization can be expressed as a linear product between the reward vector and the state occupancy. In addition, in this work we will formulate diversity maximization via an objective that is a nonlinear function of the state occupancy. While it might seem unclear which reward should be maximized to solve such an objective, we take inspiration from Convex MDPs (Zahavy et al., 2021b) where one such reward is the gradient of the objective with respect to $d_\pi$.

We begin with some formal definitions. In RL an agent interacts with an environment and seeks to maximize its cumulative reward. We consider two cases, the average reward case and the discounted case. The Markov decision process (Puterman, 1984, MDP) is defined by the tuple $(S, A, P, R)$ for the average reward case and by the tuple $(S, A, P, R, \gamma, D_0)$ for the discounted case. We assume an infinite horizon, finite state-action problem. Initially, the state of the agent is sampled according to $s_0 \sim D_0$. At time $t$, given state $s_t$, the agent selects action $a_t$ according to its policy $\pi(s_t, \cdot)$,

receives a reward $r_t \sim R(s_t, a_t)$ and transitions to a new state $s_{t+1} \sim P(\cdot, s_t, a_t)$. We consider two performance metrics, given by $v_\pi^{\mathrm{avg}} = \lim_{T \to \infty} \frac{1}{T} \mathbb{E} \sum_{t=1}^{T} r_t$, $v_\pi^\gamma = (1-\gamma) \mathbb{E} \sum_{t=1}^{\infty} \gamma^t r_t$, for the average reward case and discounted case respectively. The goal of the agent is to find a policy that maximizes $v_\pi^{\mathrm{avg}}$ or $v_\pi^\gamma$. Let $\mathbb{P}_\pi(s_t = \cdot)$ be the probability measure over states at time $t$ under policy $\pi$, then the **state occupancy** measure $d_\pi$ is given as $d_\pi^{\mathrm{avg}}(s, a) = \lim_{T \to \infty} \frac{1}{T} \mathbb{E} \sum_{t=1}^{T} \mathbb{P}_\pi(s_t = s) \pi(s, a)$, and $d_\pi^\gamma(s, a) = (1 - \gamma) \mathbb{E} \sum_{t=1}^{\infty} \gamma^t \mathbb{P}_\pi(s_t = s) \pi(s, a)$, for the average reward case and the discounted case respectively. With these, we can rewrite the RL objective in as a linear function of the occupancy measure $\max_{d_\pi \in \mathcal{K}} \sum_{s,a} r(s, a) d_\pi(s, a)$, where $\mathcal{K}$ is the set of admissible distributions (Zahavy et al., 2021b). Next, consider an objective of the form:

$$\min_{d_\pi \in \mathcal{K}} f(d_\pi), \tag{1}$$

where $f : \mathcal{K} \to \mathbb{R}$ is a nonlinear function. Sequential decision making problems that take this form include Apprenticeship Learning (AL) and pure exploration, among others (see Appendix A). In the remainder of this section, we briefly explain how to solve Eq. (1) using RL methods when the function $f$ is convex. We begin with rewriting Eq. (1) using Fenchel duality as

$$\min_{d_\pi \in \mathcal{K}} f(d_\pi) = \max_{\lambda \in \Lambda} \min_{d_\pi \in \mathcal{K}} (\lambda \cdot d_\pi - f^*(\lambda)) \tag{2}$$

where $\Lambda$ is the closure of (sub-)gradient space $\{\partial f(d_\pi) | d_\pi \in \mathcal{K}\}$, which is compact and convex (Abernethy & Wang, 2017), and $f^*$ is the Fenchel conjugate of the function $f$.

Eq. (2) presents a zero-sum max-min game between two players, the policy player $\mathrm{Alg}_\pi$ and the cost player $\mathrm{Alg}_\lambda$. We can see that from the perspective of the policy player, the objective is a linear minimization problem in $d_\pi$. Thus, intuitively speaking, the goal of the policy player is to maximize the negative cost as a reward $r = -\lambda$. To solve Eq. (2), we employ the meta algorithm from (Abernethy & Wang, 2017), which uses two online learning algorithms. The policy player $\mathrm{Alg}_\pi$ generates a *sequence* of policies $\{\pi^k\}_{k \in \mathbb{N}}$ by maximizing a sequence of negative costs $\{-\lambda^k\}_{k \in \mathbb{N}}$ as rewards that are produced by the cost player $\mathrm{Alg}_\lambda$. In this paper, the policy player uses an online RL algorithm and the cost player uses the Follow the Leader (FTL) algorithm. This implies that the cost at time $k$ is given as

$$\lambda^k = \nabla f(\bar{d}_\pi^{k-1}). \tag{3}$$

In other words, to solve an RL problem with a convex objective function (Eq. (1)), the policy player maximizes a non stationary reward that at time $k$ corresponds to the negative gradient of the objective function $f$ w.r.t $\bar{d}_\pi^{k-1}$. When the function $f$ is convex, it is guaranteed that the average state occupancy of these polices, $\bar{d}_\pi^K = \frac{1}{K} \sum_{k=1}^{K} d_\pi^k$, converges to an optimal solution to Eq. (1), *i.e.*, $\bar{d}_\pi^K \to d_\pi^* \in \arg\min_{d_\pi \in \mathcal{K}} f(d_\pi)$ (Zahavy et al., 2021b).

**Features and expected features.** We focus on the case where each state-action pair is associated with some observable features $\phi(s, a) \in \mathbb{R}^d$. For example, in the DM control suite (Tassa et al., 2018), these features correspond to the positions and velocities of the body joints being controlled by the agent. In other cases, we can learn $\phi$ with a neural network. Similar to value functions, which represent the expectation of the reward under the state occupancy, we define *expected features* as $\psi_\pi(s, a) = \mathbb{E}_{s', a' \sim d_\pi(s, a)} \phi(s', a') \in \mathbb{R}^d$. Note that in the special case of one-hot feature vectors, the expected features coincide with the state occupancy. The definition of $\psi_\pi$ depends on the state occupancy we consider. In the discounted case, $\psi_\pi^\gamma \in \mathbb{R}^d$ is also known as *Successor Features* (SFs) as defined in (Barreto et al., 2017b; 2020). In the average case, $\psi_\pi^{\mathrm{avg}} \in \mathbb{R}^d$ represents the expected features under the policy's stationary distribution and therefore it has the same value for all the state action pairs. Similar definitions were suggested in (Mehta et al., 2008; Zahavy et al., 2020b).

## 3 DISCOVERING DIVERSE NEAR-OPTIMAL POLICIES

We now introduce **D**iversity **O**ptimization **Ma**intaining **N**ear **O**ptimality, or, **DOMiNO**, which discovers a set of $n$ policies $\Pi^n = \{\pi^i\}_{i=1}^n$ by solving the optimization problem:

$$\max_{\Pi^n} \ \mathrm{Diversity}(\Pi^n) \ \mathrm{s.t} \ d_\pi \cdot r_e \geq \alpha v_e^*, \ \forall \pi \in \Pi^n, \tag{4}$$

where $v_e^*$ is the value of the optimal policy. In other words, we are looking for a set of policies that are as diverse from each other as possible, defined as $\mathrm{Diversity} : \{\mathbb{R}^{|S||A|}\}^n \to \mathbb{R}$. In addition,

we constrain the policies in the set to be nearly optimal. To define near-optimality we introduce a hyperparameter $\alpha \in [0, 1]$, such that a policy is said to be near optimal if it achieves a value that is at least $\alpha v_e^*$. In practice, we "fix" the Lagrange multiplier for the first policy $\mu^1 = 1$, so this policy only receives extrinsic reward, and use the value of this policy to estimate $v_e^*$ ($v_e^* = v_e^1$)[1]. Notice that this estimate is changing through training.

Before we dive into the details, we briefly explain the main components of DOMiNO. Building on Section 2 and, in particular, Eq. (3), we find policies that maximize the diversity objective by maximizing its gradient as a reward signal, *i.e.*, $r_d^i = \nabla_{d_\pi^i} \text{Diversity}(d_\pi^1, \ldots, d_\pi^n)$.

We propose the first candidate for this objective and derive an analytical formula for the associated reward in Eq. (5). We then move to investigate mechanisms for balancing the quality-diversity trade-off. In Eq. (7) we extend the first objective and combine it with a second, attractive force (Eq. (7)), taking inspiration from the *Van Der Waals* (VDW) force. The manner in which we combine these two forces allows us to control the degree of diversity in the set. In Section 3.1 and Section 3.2 we investigate other mechanisms for balancing the QD trade-off. We look at methods that combine the two rewards via coefficients $c_e^i$ and $c_d^i$ such that each of the policies, $\pi^1, \ldots, \pi^n$, is maximizing a reward $r^i$ that is a linear combination of the *extrinsic* reward $r_e$ and *intrinsic* reward $r_d^i$ : *i.e.*, $r^i(s, a) = c_e^i r_e(s, a) + c_d^i r_d^i(s, a)$. In Section 3.1 we focus on the method of Lagrange multipliers, which adapts the coefficients online in order to solve Eq. (4) and compare it with other multi-objective baselines in Section 3.2.

**A repulsive force**. We now present an objective that motivates policies to visit different states on average. It does so by leveraging the information about the policies' long-term behavior available in their expected features, and motivating the state occupancies to be different from each other. For that reason, we refer to this objective as a *repulsive* force (Eq. (5)).

How do we compute a set of policies with maximal distances between their expected features? To answer this question, we first consider the simpler scenario where there are only two policies in the set and consider the following objective $\max_{\pi^1, \pi^2} ||\psi^1 - \psi^2||_2^2$. This objective is related to the objective of Apprenticeship Learning (AL; Abbeel & Ng, 2004), *i.e.*, solving the problem $\min_\psi ||\psi - \psi^E||_2^2$, where $\psi^E$ are the feature expectations of an expert. Both problems use the euclidean norm in the feature expectation space to measure distances between policies. Since we are interested in diversity, we are maximizing this objective, while AL aims to minimize it.

Next, we investigate how to measure the distance of a policy from the set of multiple policies, $\Pi^n$. First we introduce the *Hausdorff* distance (Rockafellar, 1970) that measures how far two subsets $D, C$ of a metric space are from each other: $\text{Dist}(D, C) = \min_{c \in C, d \in D} ||c - d||_2^2$. In other words, two sets are far from each other in the Hausdorff distance if every point of either set is far from all the points of the other set. Building on this definition, we can define the distance from an expected features vector $\psi^i$ to the set of the other expected features vectors as $\min_{j \neq i} ||\psi^i - \psi^j||_2^2$. This equation gives us the distance between each individual policy and the other policies in the set. Maximizing it across the policies in the set, gives us our first diversity objective:

$$\max_{d_\pi^1, \ldots, d_\pi^n} 0.5 \sum_{i=1}^n \min_{j \neq i} ||\psi^i - \psi^j||_2^2. \tag{5}$$

In order to compute the associated diversity reward, we compute the gradient $r_d^i = \nabla_{d_\pi^i} \text{Diversity}(d_\pi^1, \ldots, d_\pi^n)$. To do so, we begin with a simpler case where there are only two policies, *i.e.*, $r = \nabla_{d_\pi^1} ||\psi^1 - \psi^2||_2^2 = \nabla_{d_\pi^1} ||\mathbb{E}_{s', a' \sim d_\pi^1(s,a)} \phi(s, a) - \mathbb{E}_{s', a' \sim d_\pi^2(s,a)} \phi(s, a)||_2^2 = \phi \cdot (\psi^1 - \psi^2)$, such that $r(s, a) = \phi(s, a) \cdot (\psi^1 - \psi^2)$. This reward was first derived by Abbeel & Ng (2004), but here it is with an opposite sign since we care about maximizing it. Lastly, for a given policy $\pi^i$, we define by $j_i^*$ the index of the policy with the closest expected features to $\pi^i$, *i.e.*, $j_i^* = \arg\min_{j \neq i} ||\psi^i - \psi^j||_2^2$. Using the definition of $j_i^*$, we get[2] that $\nabla_{d_\pi^i} \min_{j \neq i} ||\psi^i - \psi^j||_2^2 = \nabla_{d_\pi^i} ||\psi^i - \psi^{j_i^*}||_2^2$, and that

$$r_d^i(s, a) = \phi(s, a) \cdot (\psi^i - \psi^{j_i^*}). \tag{6}$$

---

[1]We also experimented with a variation where $v_e^* = \max_i v_e^i$. It performed roughly the same which makes sense since for most of the time $\max_i v_e^i = v_e^0$ (as policy 0 only maximizes extrinsic reward).

[2]In the rare case that the $\arg\min$ has more than one solution, the gradient is not defined, but we can still use Eq. (6) as a reward.

**The Van Der Waals force.** Next, we propose a second diversity objective that allows us to control the degree of diversity in the set via a hyperparameter $\ell_0$. As we will soon see, once a set of policies will satisfy a diversity degree of $\ell_0$, the intrinsic reward will be zero, allowing the policies to focus on maximizing the extrinsic reward (similar to a clipping mechanism). The objective is inspired from molecular physics, and specifically, by how atoms in a crystal lattice self-organize themselves at equal distances from each other. This phenomenon is typically explained as an equilibrium between two distance dependent forces operating between the atoms known as the Van Der Waals (VDW) forces; one force that is attractive and another that is repulsive.

The VDW force is typically characterized by a distance in which the combined force becomes repulsive rather than attractive (see, for example, (Singh, 2016)). This distance is called the VDW contact distance, and we denote it by $\ell_0$. In addition, we denote by $\ell_i = ||\psi^i - \psi^{j_i^*}||_2$ the Hausdorff distance for policy $i$. With this notation, we define our second diversity objective as

$$\max_{d_\pi^1,\ldots,d_\pi^n} \sum_{i=1}^n \underbrace{0.5\ell_i^2}_{\text{Repulsive}} \underbrace{-0.2\big(\ell_i^5/\ell_0^3\big)}_{\text{Attractive}}. \tag{7}$$

We can see that Eq. (7) is a polynomial in $\ell_i$, composed of two forces with opposite signs and different powers. The different powers determine when each force dominates the other. For example, when the expected features are close to each other ($\ell_i << \ell_0$), the repulsive force dominates, and when ($\ell_i >> \ell_0$) the attractive force dominates. The gradient (and hence, the reward) is given by

$$r_d^i(s,a) = (1 - (\ell_i/\ell_0)^3)\phi(s,a) \cdot (\psi^i - \psi^{j_i^*}). \tag{8}$$

We note that the coefficients in Eq. (7) are chosen to simplify the reward in Eq. (8). I.e., since the reward is the gradient of the objective, after differentiation the coefficients are 1 in Eq. (8). Inspecting Eq. (8) we can see that when the expected features are organized at the VDW contact distance $\ell_0$, the objective is maximized and the gradient is zero. We note that the powers (and the coefficients) of the attractive and repulsive forces in Eq. (7) were chosen to give a simple expression for the reward in Eq. (8) but one could consider other combinations of powers and coefficients.

### 3.1 CONSTRAINED MDPs

At the core of our approach is a solution to the CMDP in Eq. (4). There exist different methods for solving CMDPs and we refer the reader to (Altman, 1999) and (Szepesvári, 2020) for treatments of the subject at different levels of abstraction. In this work we will focus on a reduction of CMDPs to MDPs via gradient updates, known as Lagrangian methods (Borkar, 2005).

Most of the literature on CMDPs has focused on linear objectives and linear constraints. In Section 2, we discussed how to solve an unconstrained convex RL problem of the form of Eq. (1) as a saddle point problem. We now extend these results to the case where the objective is convex and the constraint is linear, i.e. $\min_{d_\pi \in \mathcal{K}} f(d_\pi)$, subject to $g(d_\pi) \leq 0$, where $f$ denotes the diversity objective and $g$ is a linear function of the form $g(d_\pi) = \alpha v_e^* - d_\pi \cdot r_e$ defining the constraint. Solving this problem is equivalent to solving the following problem:

$$\min_{d_\pi \in \mathcal{K}} \max_{\mu \geq 0} \quad f(d_\pi) + \mu g(d_\pi) = \min_{d_\pi \in \mathcal{K}} \max_{\mu \geq 0, \lambda} \lambda \cdot d_\pi - f^*(\lambda) + \mu(\alpha v_e^* - d_\pi \cdot r_e), \tag{9}$$

where the equality follows from Fenchel duality as before. Similar to Section 2, we use the FTL algorithm for the $\lambda$ player (Eq. (3)). This implies that the cost at iteration $k$, $\lambda^k$, is equivalent to the gradient of the diversity objective, which we denote by $r_d$. Eq. (9) involves a vector, $\lambda - \mu r_e$, linearly interacting with $d_\pi$. Thus, intuitively speaking, minimizing Eq. (9) from the perspective of the policy player is equivalent to maximizing a reward $r_d + \mu r_e$.

The objective for the Lagrange multiplier $\mu$ is to maximize Eq. (9), or equivalently $\mu(\alpha v_e^* - d_\pi \cdot r_e)$. Intuitively speaking, when the policy achieves an extrinsic value that satisfies the constraint, the Lagrange multiplier $\mu$ decreases (putting a smaller weight on the extrinsic component of the reward) and it increases otherwise. More formally, we can solve the problem in Eq. (9) as a *three*-player game. In this case the policy player controls $d_\pi$ as before, the cost player chooses $\lambda$ using Eq. (3), and the Lagrange player chooses $\mu$ with gradient descent. Proving this statement is out of the scope of this work, but we shall investigate it empirically.

## 3.2 MULTI-OBJECTIVE ALTERNATIVES

We conclude this section by discussing two alternative approaches for balancing the QD trade-off, which we later compare empirically with the CMDP approach. First, consider a **linear combination** method that combines the diversity objective with the extrinsic reward as

$$\max_{\Pi^n} \quad c_e d_\pi^i \cdot r_e + c_d \text{Diversity}(d_\pi^1, \ldots, d_\pi^n), \tag{10}$$

where $c_e, c_d$ are fixed weights that balance the diversity objective and the extrinsic reward. We note that the solution of such a linear combination MDP cannot be a solution to a CMDP in general. I.e., it is not possible to find the optimal dual variables $\mu^*$, plug them into Eq. (9) and simply solve the resulting (unconstrained) MDP. Such an approach ignores the fact that the dual variables must be a 'best-response' to the policy and is referred to as the "scalarization fallacy" in (Szepesvári, 2020). We now outline a few potential advantages for using CMDPs. First, the CMDP formulation guarantees that the policies that we find are near optimal (satisfy the constraint). Secondly, the weighting coefficient in linear combination MDPs has to be tuned, where in CMDPs it is adapted. This is particularly important in the context of maximizing diversity while satisfying reward. Next, consider a **hybrid approach** that combines a linear combination MDP with a CMDP as

$$\max_{\Pi^n} \quad \max(0, \alpha v_e^* - d_\pi^i \cdot r_e) + c_d \text{Diversity}(d_\pi^1, \ldots, d_\pi^n).$$

We denote by $I^i$ an indicator function for the event in which the constraint is not satisfied for policy $\pi^i$, *i.e.*, $I^i = 1$ if $d_\pi^i \cdot r_e < \alpha v_e^*$, and 0, otherwise. With this notation the reward is given by **Reverse SMERL:** $r^i(s, a) = I^i r_e(s, a) + c_d r_d^i(s, a)$ In other words, the agent maximizes a weighted combination of the extrinsic reward and the diversity reward when the constraint is violated and only the diversity reward when the constraint is satisfied. Kumar et al. (2020) proposed a similar approach where the agent maximizes a weighted combination of the rewards when the constraint is satisfied, and only the extrinsic reward otherwise: **SMERL:** $r^i(s, a) = r_e(s, a) + c_d(1 - I^i)r_d^i(s, a)$ Note that these methods come with an additional hyperparameter $c_d$ which balances the two objectives as a linear combination MDP, in addition to the optimality ratio $\alpha$.

## 4 EXPERIMENTS

Our experiments are designed to validate and get confidence in the DOMiNO agent. We emphasize that we do not explicitly compare DOMiNO with previous work nor argue that one works better than the other. Instead, we address the following questions: **(a)** Can DOMiNO discover diverse policies that are near optimal? see Fig. 2, Appendix C.1, Fig. 1b and the videos in the supplementary. **(b)** Can DOMiNO balance the QD trade-off? see Fig. 2, Fig. 2 & 3. **(c)** Do the discovered policies enable robustness and fast adaptation to perturbations of the environment? (see Fig. 4).

**Environment.** We conducted most of our experiments on domains from the DM Control Suite (Tassa et al., 2018), standard continuous control locomotion tasks where diverse near-optimal policies should naturally correspond to different gaits. Due to space considerations, we present Control Suite results on the walker.stand task. In the supplementary, however, we present similar results for walker.walk and BiPedal walker from OpenAI Gym (Brockman et al., 2016) suggesting that the method generalizes across different reward functions and domains. We also include the challenging *dog* domain with 38 actions and a $223-$dimensional observation space, which is one of the more challenging domains in control suite.

**Agent.** Fig. 1a shows an overview of DOMiNO's components and their interactions, instantiated in an actor-critic agent. While acting, the agent samples a new latent variable $z \in [1, n]$ uniformly at random at the start of each episode. We train all the policies *simultaneously* and provide this latent variable as an input. For the average reward state occupancy, the agent keeps an empirical running average for each latent policy of the rewards $\tilde{v}_{\pi^i}^{\text{avg}}$ and features (either from the environment $\phi_{obs}$ or torso embedding $\phi_{embedding}$) $\tilde{\psi}_{\pi^i}^{\text{avg}}$ encountered, where the average is taken as $\tilde{x}^i = \alpha_d \tilde{x}^{i-1} + (1 - \alpha_d)\frac{1}{T}\sum_{t=1}^{T} x^i(s_t, a_t)$ with decay factor $\alpha_d$. Varying $\alpha_d$ can make the estimation more online (small $\alpha_d$, as used for the constraint), or less online (large $\alpha_d$, as needed for Eq. (4)). The agent uses $\tilde{\psi}_{\pi^i}^{\text{avg}}$ to compute the diversity reward as described in Eq. (6). $\tilde{v}_{\pi^i}^{\text{avg}}$ is used to optimize the Lagrange multiplier $\mu$ for each policy as in Eq. (9) which is then used to weight the quality and diversity advantages for the policy gradient update. Pseudo code and further implementation details, as well as treatment of the discounted state occupancy, can be found in Appendix B.

**Quality and diversity.** To measure diversity qualitatively, we present "motion figures" by discretizing the videos (details in the Appendix) that give a fair impression of the behaviors. The videos, associated with these figures, can be found in the supplementary as well. Fig. 1b presents ten polices discovered by DOMiNO with the repulsive objective and the optimality ratio set to $0.9$. The policies are ordered from top-left to bottom right, so policy 1, which only maximizes extrinsic reward and sets the constraint, is always at the top left. The policies exhibit different types of standing: standing on both legs, standing on either leg, lifting the other leg forward and backward, spreading the legs and stamping. Not only are the policies different from each other, they also achieve high extrinsic reward in standing (see values on top of each policy visualization). Similar figures for other domains can be found in Appendix C.1.

To further study the QD trade-off, we use scatter plots, showing the episode return on the y-axis, and the diversity score, corresponding to the Hausdorff distance (Eq. (5)), on the x-axis. The top-right corner of the diagram, therefore, represents the most diverse and highest quality policies. Each figure presents a sweep over one or two hyperparameters and we use the color and a marker to indicate the values. In all of our experiments, we report $95\%$ confidence intervals. In the scatter plots, they correspond to 5 seeds and are indicated by the crosses surrounding each point.

Fig. 2 (left) presents the results for DOMiNO with the repulsive reward in the walker.stand domain. We can observe that regardless of the set size, DOMiNO achieves roughly the same extrinsic reward across different optimality ratios (points with the same color obtain the same y-value). This implies that the constraint mechanism is working as expected across different set sizes and optimality ratios. In addition, we can inspect how the QD trade-off is affected when changing the optimality ratio $\alpha$ for sets of the same size (indicated in the figures with light lines). This observation can be explained by the fact that for lower values of $\alpha$, the volume of the constrained set is larger, and therefore, it is possible to find more diversity within it. This behavior is consistent across different set sizes, though it is naturally more difficult to find a set of diverse policies as the set size gets larger (remember that we measure the distance to the closest policy in the set).

We present the same investigation for the **VDW** reward in Fig. 2 (center). Similar to the repulsive reward, we can observe that the constraint is satisfied, and that reducing the optimality ratio allows for more diversity. Fig. 2 (right) shows how different values of $\ell_0$ affect the QD trade-off for a set of size 10. We can observe that the different combinations of $\ell_0$ and $\alpha$ are organized as a grid in the QD scatter, suggesting that we can control both the level of optimality and the degree of diversity by setting these two interpretable hyperparameters. Lastly, Fig. C8 in the end of Appendix C.2 presents results for an additional experiment where we use only the VDW reward without the constraints.

Fig. 3 compares the QD balance yielded by DOMiNO to the alternative strategies described in Section 3.2. Specifically, we look at DOMiNO's Lagrangian method, the linear combination of the objectives (Eq. (10)), and the two hybrid strategies, SMERL and Reverse SMERL for a set of ten policies in walker.stand. Note that in all cases we are using DOMiNO's repulsive diversity objective, and the comparison is strictly about strategies for combining the quality and diversity objectives. The plot for each strategy shows how the solution to the QD tradeoff varies according to the relevant hyperparameters for that strategy, namely, the optimality ratio $\alpha$ for DOMiNO, the fixed constant $c_e$ for the linear combination strategy (we implicitly set $c_d = 1 - c_e$), and both $\alpha$ and constant $c_d$ for the hybrid approaches (in the hybrid plots, $c_d$ value is labeled as a marker, while $\alpha$ is indicated by color).

For the linear combination, shown on the right, the $c_e$ parameter proves ill-behaved and choppy, quickly jumping from the extreme of all diversity no quality to the opposite, without a smooth interim.

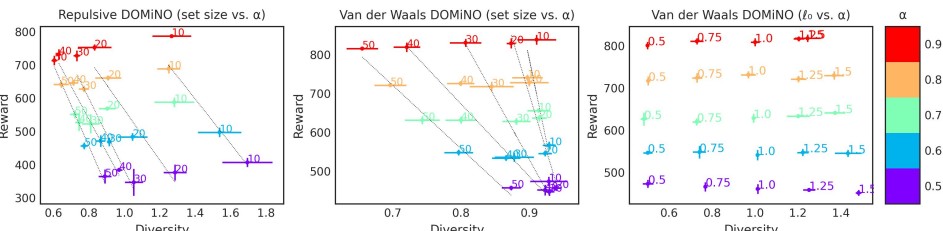

Figure 2: DOMiNO QD results in walker.stand. **Left**: Set size vs. optimality ratio ($\alpha$) with the repulsive reward. **Center**: Set size vs. $\alpha$ with the VDW reward ($\ell_0 = 1$). **Right**: Target diversity ($\ell_0$) vs. $\alpha$ with the VDW reward.

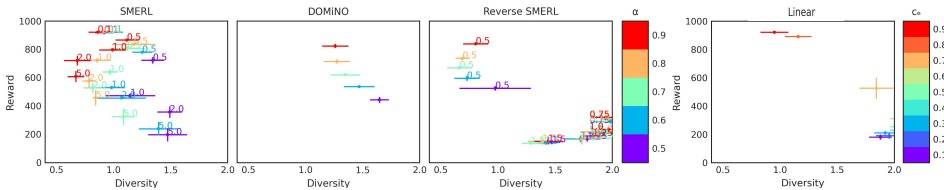

Figure 3: DOMiNO's Lagrangian method finds only solutions that push the upper right boundary of quality and diversity, and varies in a smooth, interpretable way with its only hyperparameter, $\alpha$, contrasted with the jumpy nature of the linear combination hyperparameter $c_e$, and the redundancy of the hyperparameters in the hybrid methods.

In contrast, the DOMiNO approach of solving the CMDP directly for the Lagrange multiplier yields solutions that push along the upper right diagonal boundary, finding the highest diversity (farthest right) set of policies for a given optimality ratio (color), varying smoothly along this line as $\alpha$ varies. Another advantage of DOMiNO's approach is that it *only* finds such QD-optimal solutions, where, in contrast, SMERL (left), when appropriately tuned, can also yield some solutions along the upper-right QD border, but often finds sub-optimal solutions, and therefore must be tuned further with $c_d$ to find the best solutions. We further explore the difficulty tuning SMERL in the supplementary (Fig. C6) and find that the best $c_d$ for 10 policies provides solutions with no diversity for other set sizes.

**Feature analysis** The choice of feature space used for optimizing diversity can have a huge impact on the kind of diverse behavior learned. In environments where the observation space is high dimensional and less structured (e.g. pixel observations), computing diversity using the raw features may not lead to meaningful behavior. As specified in Section 3 the feature space used to compute diversity in our Control Suite experiments throughout the paper corresponds to the positions and velocities of the body joints returned as observations by the environment. We show that it is feasible to use a learned embedding space instead. As a proof of principle we use the output of the torso network as a learned embedding described in Section 4 for computing our diversity metric.

Table 1 compares the diversity measured in raw observation features (first row) and embedding features (second row) in the walker.stand domain. Columns indicate the feature space that was used to

|  | $\phi_{\text{obs}}$ | $\phi_{\text{embedding}}$ |
|---|---|---|
| Observation | $1.21 \pm 0.05$ | $1.01 \pm 0.05$ |
| Embedding | $2.14 \pm 0.09$ | $2.35 \pm 0.09$ |

Table 1

compute the diversity objective during training averaged across 20 seeds. Inspecting the table, we can see that agents trained to optimize diversity in the learned embedding space and agents that directly optimize diversity in the observation space achieve comparable diversity if measured in either space, indicating that learned embeddings can feasibly be used to achieve meaningful diversity.

**Closing the loop: $k$-shot adaptation.** We motivated qualitative diversity by saying that diverse solutions can be robust and allow for rapid adaptation to new tasks and changes in the environment. Here we validate this claim in a k-shot adaptation experiment: we train a set of diverse high quality policies on a canonical benchmark task, then test their ability to adapt to environment and agent perturbations. These include four kinematics and dynamics perturbations from the Real World RL suite (Dulac-Arnold et al., 2019), and a fifth perturbation inspired by a "motor failure" condition (Kumar et al., 2020) which, every 50 steps and starting at $T = 10$, disables action-inputs for the first two joints for a fixed amount of time. In Fig. 4, we present the results in the walker.walk and walker.stand domains (rows). Columns correspond to perturbation types and the x-axis corresponds to the perturbation.

**K-shot** adaptation is measured in the following manner. For each perturbed environment, and each method, we first execute $k = 10$ environment trajectories with each policy. Then, for each method we select the policy that performs the best in the set. We then evaluate this policy for 40 more trajectories and measure the average reward of the selected policy $r_{\text{method}}$. The y-axis in each figure measures $r_{\text{method}}/r_{\text{baseline}}$, where $r_{\text{baseline}}$ measures the reward in the perturbed environment of an RL baseline agent that was trained with a single policy to maximize the extrinsic reward in the original task. We note that the baseline was trained with the same RL algorithm, but without diversity, and it matches the state-of-the-art in each training domain (it is almost optimal). We found the relative metric to be the most informative overall, but it might be misleading in situations where all the methods perform roughly the same (e.g. in Fig. 4 bottom right). For that reason, we also include a figure with the raw rewards $r_{\text{method}}, r_{\text{baseline}}$ in the supplementary (Fig. E10). Lastly, we repeat this process across 20 training seeds, and report the average with a 95% Confidence Interval (CI).

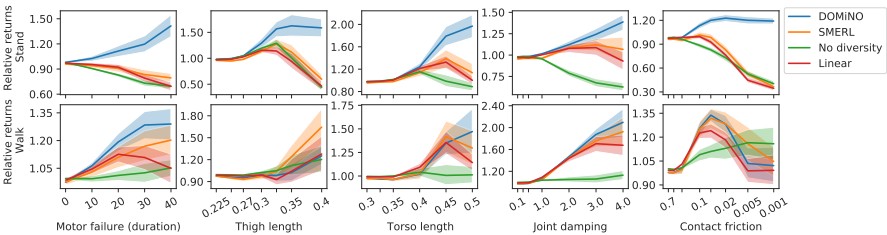

Figure 4: K-shot adaptation in Control Suite. We report mean episode return (95% CI) on held-out test tasks relative to the performance of a single policy trained on extrinsic rewards. While not invariant to sudden changes in the environment, DOMiNO is more robust to a variety of perturbations.

We compare the following methods: DOMiNO, SMERL, Linear Combination, and No diversity, where all the diversity methods use the diversity reward from Eq. (6) with 10 policies. Since we are treating the perturbed environments as hold out tasks, we selected the hyper parameters for each method based on the results in Fig. 3, *i.e.* we chose the configuration that was the most qualitatively diverse (in the upper-right most corner of Fig. 3). Concretely, for DOMiNO and SMERL $\alpha = 0.9$, for SMERL $c_d = 0.5$ and for linear combination $c_e = 0.7$. More K-shot adaptation curves with other hyper parameter values can be found in Appendix E. The No diversity method is similar to $r_{\text{baseline}}$, but uses 10 policies that all maximize the extrinsic reward (instead of a single policy).

Inspecting Fig. 4 we can see that for small perturbations, DOMiNO retains the performance of the baseline. However, as the magnitude of the perturbation increases, the performance of DOMiNO is much higher than the baseline (by a factor of $1.5 - 2.0$). This observation highlights that a diverse set of policies as found by DOMiNO is much more capable at handling changes to the environment and can serve as a strong starting point for recovering optimal behavior. As we have shown in 3.2, other approaches to managing the trade-off between quality and diversity such as SMERL are much more sensitive to the choice of hyper-parameters and require significant tuning. While SMERL is able to find a useful, diverse set of policies with some effort, it is difficult to match DOMiNO's performance across all pertubations and tasks. See the supplementary material for further comparison of DOMiNO with SMERL and linear combination over more hyper parameters. We also include a video that illustrates how the individual policies adapt to the environment perturbations.

## 5 CONCLUSIONS

In this work we proposed DOMiNO, an algorithm for discovering diverse behaviors that maintain optimality. We framed the problem as a CMDP in the state occupancies of the policies in the set and developed an end-to-end differentiable solution to it based on reward maximization.

We note that the diversity maximization objectives we explored are not convex RL problems; the opposite problem of minimizing the distance between state occupancies that is often used for Apprenticeship Learning is convex. Nevertheless, we explored empirically the application of convex RL algorithms to the non convex problem of maximizing diversity. In our experiments we demonstrated that the policies discovered by DOMiNO, or, *DOMiNO's* $\pi$, are diverse and maintain optimality. We then explored how DOMiNO balances the QD trade-off and compared it with linear combination baselines. Our results suggest that DOMiNO can control the degree of quality and diversity via interpretable hyperparameters, while other baselines struggle to capture both.

In particular, in Fig. 2 (right) we showed that DOMiNO indeed solves the problem it is designed to solve: it satisfies the constraint and achieves the diversity specified by the VDW reward ($\ell_0$). This is perhaps not surprising, the (non convex) optimization of Deep Neural Networks often builds on convex optimization algorithms to great success. Nevertheless we believe that our results make an important demonstration of this concept to the study of diverse skill discovery.

In our K-shot experiments we demonstrated that DOMiNO's $\pi$ can adapt to changes in the environment. An exciting direction for future work is to use DOMiNO in a never ending RL setting, where the environment changes smoothly over time, and see if maintaining a set of QD diverse policies will make it more resilient to such changes.

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

APPENDIX

## A  RELATED WORK

**QD**. Quality-Diversity optimization is a type of evolutionary algorithm that aims at generating large collections of diverse solutions that are all high-performing (Pugh et al., 2016; Cully & Demiris, 2017, QD). It comprises two main families of approaches: MAP-Elites (Cully et al., 2015; Mouret & Clune, 2015) and novelty search with local competition (Lehman & Stanley, 2011, NSLC), which both distinguish and maintain policies that are different in the behaviour space. The main difference is how the policy collection is implemented, either as a structured grid or an unstructured collection, respectively. The behaviour space is either handcrafted based on domain knowledge (Cully et al., 2015; Tarapore et al., 2016) or learned from data (Cully, 2019). Some other works combine evolutionary QD with RL (Pierrot et al., 2022; Tjanaka et al., 2022; Nilsson & Cully, 2021). Further references can be found on the QD webpage.

Our paper share its goals with those of the QD formulation, but it differs from this line of work in the type of optimization used to find the diverse high quality policies; while QD algorithms focus on evolution, our approach is based on reward maximization. This is feasible thanks to the introduction of Lagrange multipliers and because we apply the Fenchel transform to the diversity objective as we explained in Section 2.

**MDPs with general objectives**. Our work builds on recent advancements in the study of MDPs with an objective that is convex in the state occupancy and includes Apprenticeship Learning (Abbeel & Ng, 2004; Zahavy et al., 2020a; Shani et al., 2021; Belogolovsky et al., 2021) and maximum state entropy exploration (Hazan et al., 2019). Recently, a few papers were published on solving general convex MDPs (Hazan et al., 2019; Zhang et al., 2020; Geist et al., 2021; Zahavy et al., 2021b; Mutti et al., 2022).

Our work considers an objective that is not convex but uses the same algorithms that were developed for the convex case as we explained in Section 2. It might surprise some readers that the same algorithms perform well in the non convex case, nevertheless this is what we found empirically. We also note that it is also the case that convex optimization algorithms are found to be useful for optimizing deep neural nets, which are not convex. That said, we hope to further consider the implications of optimizing non convex MDPs with Convex MDP techniques in future work.

**Intrinsic rewards**. Intrinsic rewards have been used for discovering diverse skills. The most common approach is to define diversity in terms of the discriminability of different trajectory-specific quantities and to use these ideas to maximize the Mutual information between states and skills (Gregor et al., 2017; Eysenbach et al., 2019; Sharma et al., 2020; Baumli et al., 2021). Other works implicitly induce diversity to learn policies that maximize the set robustness to the worst-possible reward (Kumar et al., 2020; Zahavy et al., 2021a), and other add diversity as a regularizer when maximizing the extrinsic reward (Hong et al., 2018; Masood & Doshi-Velez, 2019; Peng et al., 2020; Sun et al., 2020; Zhang et al., 2020; Sharma et al., 2020; Badia et al., 2020; Pathak et al., 2017).

It has been recently suggested that the Mutual Information (MI) between policies and states is equivalent to the average KL divergence between state occupancies (Zahavy et al., 2021b; Eysenbach et al., 2021). The MI is minimized in GAIL for AL (Ho & Ermon, 2016) and maximized for diversity in VIC and DIAYN (Eysenbach et al., 2019; Gregor et al., 2017). Our framework for modeling diversity objectives and deriving intrinsic rewards from them generalizes these results (which were specific to the mutual information) and allows one to introduce new objectives for diversity maximization and easily derive intrinsic rewards for them.

Lastly, Parker-Holder et al. (2020) measure diversity between behavioral embeddings of policies. The advantage is that the behavioral embeddings can be defined via a differential function directly on the parameters of the policy, which makes the diversity objective differentiable w.r.t to the diversity measure. Our approach is similar in the sense that state-occupancies are also behavioral embeddings; the state occupancy characterizes the policy perfectly as $\pi(s,a) = \frac{d_\pi(s,a)}{\sum_a d_\pi(s,a)}$ (in states that are visited with non zero probability). This representation is not differentiable w.r.t the policy parameters, and we overcome this hurdle by deriving a reward signal that leads to maximizing diversity in this representation.

**Using Successor Features to discover policies.** In a related line of work, Successor Features are used to discover and control policies (Tasse et al., 2021; Nangue Tasse et al., 2020; Zahavy et al., 2021a;a; Alver & Precup, 2021; Barreto et al., 2017a; 2018; 2019). In this line of work, there is a sequence of rounds, and a new policy is discovered in each round by maximizing a **stationary** reward signal (the objective for each new policy is a standard RL problem). The rewards can be random vectors, one hot vectors, or an output of an algorithm (for example, an algorithm can be designed to increase the diversity of the set).

The main difference from this line of work and our paper is that we train all the policies in parallel via a shared latent architecture. This approach has the advantage that it is more sample efficient (since we solve a single problem for all the policies and not a new problem for each policy). In addition, some of the iterative approaches can be viewed as solutions for convex MDPs. That is, it can be shown that by optimizing a sequence of policies and defining a new reward in each round as the function of the previous state occupancies , the procedure converges to a solution for a convex MDP (Abbeel & Ng, 2004; Zahavy et al., 2021a). However, in DOMiNO we train all the policies in parallel to maximize a single objective via a non stationary reward as we described in Section 2.

**Constrained MDPs and multi objective QD**. Constrained MDPs are a popular model with vast literature; we refer the reader to (Altman, 1999; Szepesvári, 2020) for more details. The most popular approach for solving CMDPs is to use Lagrange multipliers Section 3.1. The main advantage with this approach is that the CMDP is reduced to an MDP with a non stationary reward, which makes it possible to use existing RL algorithms to analalyze and implement a solution for a CMDP (Borkar, 2005; Bhatnagar & Lakshmanan, 2012; Tessler et al., 2019; Stooke et al., 2020; Calian et al., 2021; Huang et al., 2020).

A different approach for solving CMDPs is SMERL (Kumar et al., 2020) which we analyzed in Section 3.2. SMERL and the reverse SMERL variation performed quite similar to the Lagrange multipliers approach, but required to be properly tuned. Another related idea is the reward switching mechanism (Zhou et al., 2022), which switches between extrinsic and intrinsic rewards via a trajectory-based novelty measurement during the optimization process.

There are also approaches that aim to solve QD problems defined as constraint optimization problems. For example, the NOAH algorithm (Ulrich & Thiele, 2011) is an evolutionary algorithm which determines a maximally diverse set of solutions whose objective values are below a provided objective barrier. It does so by iteratively switching between objective value and set-diversity optimization while automatically adapting a constraint on the objective value until it reaches the barrier. The challenge in using this kind of algorithm in the RL setup is that it requires solving many RL problems (for different barriers). The Lagrange multipliers approach, on the other hand, allows us to optimize all the policies in parallel, together with the Lagrange multipliers, which is more computationally efficient.

In (Zhang et al., 2019) the authors propose to compute two gradients, one for the reward and one for the diversity objective, and then to update the policy in the direction of the angular bisector of the two objectives.

We also note that there had been many works that considered using a linear combination of rewards to balance the QD trade-off (Hong et al., 2018; Masood & Doshi-Velez, 2019; Parker-Holder et al., 2020; Gangwani et al., 2020; Peng et al., 2020). However, as we note before, these methods cannot find solutions to CMDPs (in general) due to the scalarization fallacy (Szepesvári, 2020). In addition, we found empirically that they do not find good solutions to the QD trade-off and instead find policy sets that are either diverse or of high quality, and do not find solutions with more flexible balance.

**Diversity in Multi Agent RL**. There has also been a lot of work in multi agent RL on discovering diverse sets of policies. In this case, the idea is typically to discover diverse high quality policies and then to train a Best Response policy that exploits all of the discovered policies. For example, Lupu et al. (2021) study zero-shot coordination. They propose train a common best response to a population of agents, which they regulate to be diverse. The diversity objective defined in terms of trajectories and use the Jensen-Shannon divergence to optimize it. Liu et al. (2017) consider diversity in Markov games. They proposed a state-occupancy based diversity objectives via the shared state-action occupancy of all the players in the game (in addition to a second response based diversity metric that is specific to multi-agent domains). They then use an empowerment based reward to increase the diversity of this state occupancy. This is a very interesting idea but somewhat orthogonal

to our work since we focus on the single agent scenario. In addition to that, our approach generalizes the class of objectives, which typically focused on mutual information, to a more general class of diversity functions and to show that there is an easy way to derive an intrinsic reward for them. We also proposed two new diversity objectives based on these ideas (the Hausdorff and the VDW). We believe that this is an exciting direction as it will allow others to propose other objectives and then simply differentiate them to get a reward.

**Repulsive forces**. In a related line of work, Vassiliades et al. (2016) suggested to use Voronoi tessellation to partition the feature space of the MAP-Elite algorithm to regions of equal size and Liu et al. (2017) proposed a Stein Variational Policy Gradient with repulsive and attractive components. Flet-Berliac et al. (2021) introduced an adversary that mimics the actor in an actor critic agent, and then added a repulsive force between the agent and the adversary. Using a VDW force to control diversity is novel to the best of our knowledge.

## B    IMPLEMENTATION DETAILS

**Distributed agent** Acting and learning are decoupled, with multiple actors gathering data in parallel from a batched stream of environments, and storing their trajectories, including the latent variable $z$, in a replay buffer and queue from which the learner can sample a mixed batch of online and replay trajectories (Schmitt et al., 2020; Hessel et al., 2021). The latent variable $z$ is sampled uniformly at random in $[1, n]$ during acting at the beginning of each new episode. The learner differentiates the loss function as described in Algorithm 1, and uses the optimizer (specified in Table 2) to update the network parameters and the Lagrange multipliers (specified in Table 3). Lastly, the learner also updates the and moving averages as described in Algorithm 1.

**Initialization** When training begins we initialize the network parameters as well as the Lagrange multipliers: $\mu^i = \sigma^{-1}(0.5), \forall i \in [2, n]$, where $\sigma^{-1}$ is the inverse of the Sigmoid function $\mu^1 = 1$; and the moving averages: $\tilde{v}_{\pi^i}^{\mathrm{avg}} = 0., \forall i \in [1, n]$, $\tilde{\psi}_{\pi^i}^{\mathrm{avg}} = \bar{1}/d, \forall i \in [1, n]$. Here $n$ is the number of policies and $d$ is the dimension of the features $\phi$.

**Bounded Lagrange multiplier** To ensure the Lagrange multiplier does not get too large so as to increase the magnitude of the extrinsic reward and destabilize learning, we use a bounded Lagrange multiplier (Stooke et al., 2020) by applying Sigmoid activation on $\mu$ so the effective reward is a convex combination of the diversity and the extrinsic rewards: $r(s, a) = \sigma(\mu^i)r_e(s, a) + (1 - \sigma(\mu^i))r_d^i(s, a)$, and the objective for $\mu$ is $\sigma(\mu)(\alpha v_e^* - d_\pi \cdot r_e)$.

**Average state occupancy** The empirical feature averages used for experiments in the main text are good, though imperfect due to the bias from samples before the policy mixes. In our experiments, however, since the mixing time for the DM Control Suite is much shorter than the episode length $T$, the bias is small ($\sim 5\%$).

**Discounted state occupancy** For a more scalable solution, as mentioned in Section 2, we can instead predict successor features using an additional network head as shown in Fig. C7a. Similar to value learning, we use V-trace (Espeholt et al., 2018) targets for training successor features. In discounted state occupancy case we also use the extrinsic value function of each policy $v_e^i$ (Fig. 1a) to estimate $d_\pi \cdot r_e$, instead of the running average $\tilde{v}_{\pi^i}^{\mathrm{avg}}$. We show experimental results for this setup in Fig. C7b.

**Loss functions.** Instead of learning a single value head for the combined reward, our network has two value heads, one for diversity reward and one for extrinsic reward.

We use V-trace (Espeholt et al., 2018) to compute td-errors and advantages for each of the value heads using the "vtrace td error and advantage" function implemented here https://github.com/deepmind/rlax/blob/master/rlax/_src/vtrace.py. The value loss for each head is the squared $\ell_2$ loss of the td-errors, and the combined value loss for the network is the sum of these two losses: $\mathrm{td}_d^2 + \mathrm{td}_e^2$.

In addition to that, our network has a policy head that is trained with a policy gradient loss as implemented in https://github.com/deepmind/rlax/blob/master/rlax/_src/policy_gradients.py). When training the policy, we combine the intrinsic and extrinsic advantages $\delta = \sigma(\mu^i)\delta_e + (1 - \sigma(\mu^i))\delta_d$ (see the Weight cumulants function in Appendix B.1) which has the same effect as combining the reward. However, we found that having two value heads is more stable as each value can have a different scale.

The final loss of the agent is a weighted sum of the value loss the policy loss and the entropy regularization loss, and the weights can be found in Table 2.

Algorithm 1 also returns a Lagrange loss function, designed to force the policies to achieve a value that is at least $\alpha$ times the value of the first policy (which only maximizes extrinsic reward), where $\alpha$ is the optimally ratio (Table 3). We update the Lagrange multipliers $\mu$ using the optimizer specified in Table 3 but keep the multiplier of the first policy fixed $\mu^1 = 1$.

Lastly, Algorithm 1 also updates the moving averages.

---

**Algorithm 1:** Loss function

---

**Parameters:** Network parameters $\theta$, Lagrange multipliers $\mu$, moving averages $\left\{\tilde{v}_{\pi^i}^{\mathrm{avg}}\right\}_{i=1}^n$, $\left\{\tilde{\psi}_{\pi^i}^{\mathrm{avg}}\right\}_{i=1}^n$.

**Data:** $m$ trajectories $\{\tau_j\}_{j=1}^m$ of size $T$, $\tau_j = \left\{z^j, x_s^j, a_s^j, r_s^j, \phi_s^j, \mu(a_s^j|x_s^j)\right\}_{s=1}^T$, where $\mu(a_s^j|x_s^j)$ is the probability assigned to $a_s^j$ in state $x_s^i$ by the behaviour policy $\mu(a|x)$.

**Forward pass:** $\{\pi(a_s^j|x_s^j), v_e(x_s^j), v_d(x_s^j)\} \leftarrow \mathrm{Network}(\{\tau_j\}_{j=1}^m)$

**Compute extrinsic td-errors and advantages:** $\mathrm{td}_e(x_s^j), \delta_e(x_s^j) \leftarrow$ V-trace with extrinsic reward $r_s^j$ and extrinsic critic $v_e(x_s^j)$

**Compute intrinsic reward:** $r_d^i(x_s^j)$ from $\tilde{\psi}_{\pi^z}^{\mathrm{avg}}, z, \phi_s^j$ with Eq. (6) or (8)

**Compute intrinsic td-errors and advantages:** $\mathrm{td}_d(x_s^j), \delta_d(x_s^j) \leftarrow$ V-trace with intrinsic reward $r_s^j$ and intrinsic critic $v_d(x_s^j)$

**Combine advantages:** $\delta(x_s^j) = \sigma(\mu^i)\delta_e(x_s^j) + (1 - \sigma(\mu^i))\delta_d(x_s^j)$

**Weighted loss:**

$$\sum_{s,j} b_v(\mathrm{td}_e(x_s^j)^2 + \mathrm{td}_d(x_s^j)^2) + b_\pi \log(\pi(a_s^j|x_s^j))\delta(x_s^j) + b_{\mathrm{Ent}}\mathrm{Entropy}(\pi(a_s^j|x_s^j))$$

**Lagrange loss:**

$$\sum_{i=1}^n \sigma(\mu^i)(\tilde{v}_{\pi^i}^{\mathrm{avg}} - \alpha\tilde{v}_{\pi^1}^{\mathrm{avg}})$$

**Update moving averages:**

$$\tilde{v}_{\pi^i}^{\mathrm{avg}} = \alpha_d^{\tilde{v}^{avg}}\tilde{v}_{\pi^i}^{\mathrm{avg}} + (1 - \alpha_d^{\tilde{v}^{avg}})r_t, \quad \tilde{\psi}_{\pi^i}^{\mathrm{avg}} = \alpha_d^{\tilde{\psi}^{avg}}\tilde{\psi}_{\pi^i}^{\mathrm{avg}} + (1 - \alpha_d^{\tilde{\psi}^{avg}})\phi_t$$

**return** *Weighted loss, Lagrange loss*, $\left\{\tilde{v}_{\pi^i}^{\mathrm{avg}}\right\}_{i=1}^n, \left\{\tilde{\psi}_{\pi^i}^{\mathrm{avg}}\right\}_{i=1}^n$

---

## B.1 FUNCTIONS

```python
def intrinsic_reward(phi, sfs, latents, attractive_power=3.,
        repulsive_power=0., attractive_coeff=0., target_d=1.):
  """Computes a diversity reward using successor features.

 Args:
      phi: features [tbf].
      sfs: avg successor features [lf] or
      predicted, discounted successor features [tbfl].
      latents: [tbl].
      attractive_power: the power of the attractive force.
      repulsive_power: the power of the repulsive force.
      attractive_coeff: convex mixing of attractive & repulsive forces
      target_d(\ell_0): desired target distance between the sfs.
      When attractive_coeff=0.5, target_d is the minimizer of the
      objective, i.e., the gradient (the reward) is zero.
  Returns:
      intrinsic_reward.
  """
  # If sfs are predicted we have 2 extra leading dims.
  if jnp.ndim(sfs) == 4:
    sfs = jnp.swapaxes(sfs, 2, 3)  # tbfl -> tblf (to match lf shape of
    avg sf)
    compute_dist_fn = jax.vmap(jax.vmap(compute_distances))
    matmul_fn = lambda x, y: jnp.einsum('tbl,tblf->tbf', x, y)
  elif jnp.ndim(sfs) == 2:
    compute_dist_fn = compute_distances
    matmul_fn = jnp.matmul
  else:
    raise ValueError('Invalid shape for argument 'sfs'.')
  l, f = sfs.shape[-2:]
  # Computes an tb lxl matrix where each row, corresponding to a latent,
    is a 1 hot vector indicating the index of the latent with the closest
     sfs
  dists = compute_dist_fn(sfs, sfs)
  dists += jnp.eye(l) * jnp.max(dists)
  nearest_latents_matrix = jax.nn.one_hot(
      jnp.argmin(dists, axis=-2), num_classes=l)
  # Computes a [tbl] vector with the nearest latent to each latent in
    latents
  nearest_latents = matmul_fn(latents, nearest_latents_matrix)
  # Compute psi_i-psi_j
  psi_diff = matmul_fn(latents - nearest_latents, sfs)  # tbf
  norm_diff = jnp.sqrt(jnp.sum(jnp.square(psi_diff), axis=-1)) / target_d
  c = (1. - attractive_coeff) * norm_diff**repulsive_power
  c -= attractive_coeff * norm_diff**attractive_power
  reward = c * jnp.sum(phi * psi_diff, axis=-1) / f
  return reward

def l2dist(x, y):
  """Returns the L2 distance between a pair of inputs."""
  return jnp.sqrt(jnp.sum(jnp.square(x - y)))

def compute_distances(x, y, dist_fn=l2dist):
  """Returns the distance between each pair of the two collections of
    inputs."""
  return jax.vmap(jax.vmap(dist_fn, (None, 0)), (0, None))(x, y)
```

Listing 1: Intrinsic Reward

```python
def weight_cumulants(lagrange, latents, extrinsic_cumulants,
    intrinsic_cumulants):
    """Weights cumulants using the Lagrange multiplier.

    Args:
      lagrange: lagrange [l].
      latents: latents [tbl].
      extrinsic_cumulants: [tb].
      intrinsic_cumulants: [tb].

    Returns:
      extrinsic reward r_e and intrinsic_reward r_d.
    """
    sig_lagrange = jax.nn.sigmoid(lagrange)   # l
    latent_sig_lagrange = jnp.matmul(latents, sig_lagrange)   # tb
    # No diversity rewards for latent 0, only maximize extrinsic reward
    intrinsic_cumulants *= (1 - latents[:, :, 0])
    return (1 - latent_sig_lagrange) * intrinsic_cumulants +
    latent_sig_lagrange * extrinsic_cumulants
```

Listing 2: Weight cumulants

```python
def lagrangian(lagrange, r, optimality_ratio):
    """Loss function for the Lagrange multiplier.

    Args:
      lagrange: lagrange [l].
      r: moving averages of reward [l].
      optimality_ratio: [l]."""
    l_ = jax.nn.sigmoid(lagrange)
    return jnp.sum(l_ * (r -  r[0] * optimality_ratio))
```

Listing 3: lagrange loss function

## B.2 Motion figures

Our "motion figures" were created in the following manner. Given a trajectory of frames that composes a video $f_1, \ldots, f_T$, we first trim and sub sample the trajectory into a point of interest in time: $f_n, \ldots, f_{n+m}$. We always use the same trimming across the same set of policies (the sub figures in a figure). We then sub sample frames from the trimmed sequence at frequency $1/p$: $f_n, f_{n+p}, f_{n+2p} \ldots,$. After that, we take the maximum over the sequence and present this "max" image. In Python for example, this simply corresponds to

```python
n=400, m=30, p=3
indices = range(n, n+m, p)
im = np.max(f[indices])
```

This creates the effect of motion in single figure since the object has higher values than the background.

## B.3 Hyperparameters

The hyperparameters in Table 2 are shared across all environments except in the BiPedal Domain the learning rate is set to $10^{-5}$ and the learner frames are $5 \times 10^7$. We report the DOMiNO specific hyperparameters in Table 3.

| Hyperparameter | Value |
|---|---|
| Replay capacity | $5 \times 10^5$ |
| Learning rate | $10^{-4}$ |
| Learner frames | $2 \times 10^7$ |
| Discount factor | 0.99 |
| $b_{\text{Ent}}$ Entropy regularization weight | 0.01 |
| $b_{\pi}$ Policy loss weight | 1.0 |
| $b_v$ Value loss weight | 1.0 |
| Replay batch size | 600 |
| Online batch size | 6 |
| Sequence length | 40 |
| Optimizer | RMSprop |

Table 2: General hyperparameters

| Hyperparameter | Control Suite | BiPedal Walker |
|---|---|---|
| $\alpha$ Optimality ratio | 0.9 | 0.7 |
| Lagrange initialization | 0.5 | 0.5 |
| Lagrange learning rate | $10^{-3}$ | $10^{-3}$ |
| Lagrange optimizer | Adam | Adam |
| $\tilde{v}_{\pi^i}^{\text{avg}}$ decay factor $\alpha_d^{\tilde{v}^{avg}}$ | 0.9 | 0.999 |
| $\tilde{\psi}_{\pi^i}^{\text{avg}}$ decay factor $\alpha_d^{\tilde{\psi}^{avg}}$ | 0.99 | 0.9999 |

Table 3: DOMiNO hyperparameters

## C  ADDITIONAL EXPERIMENT RESULTS

### C.1  MOTION FIGURES

We now present motion figures, similar to Fig. 1b, but in other domains (see Fig. 6-9). The videos, associated with these figures can be found in a separate .zip file. Each figure presents ten polices discovered by DOMiNO and their associated rewards (in white text) with the repulsive objective and the optimality ratio set to $0.9$. As we can see, the policies exhibit different gaits.

Next to each figure, we also present the distances between the expected features of the discovered policies measured by the $\ell_2$ norm. In addition, in each row $i$ we use a dark black frame to indicate the the index of the policy with the closest expected features to $\pi^i$, *i.e.*, in the i-th row we highlight the j-th column such that $j = j_i^* = \arg\min_{j \neq i} ||\psi^i - \psi^j||_2^2$.

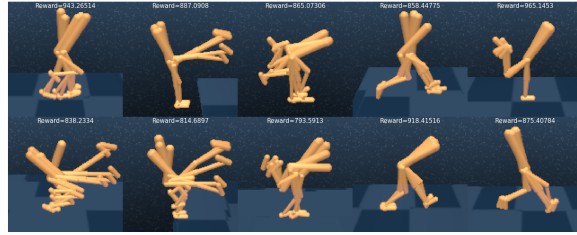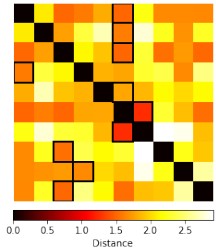

Figure C1: QD in walker.stand. Left: 10 policies and corresponding rewards. Right: distances in $\ell_2$ norm between the Successor features of the policies.

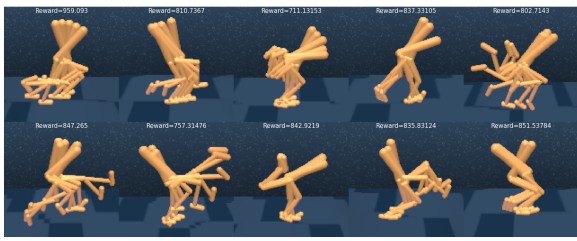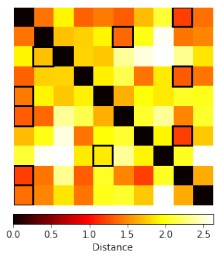

Figure C2: QD in walker.walk. **Left:** 10 policies and corresponding rewards. **Right:** distances in $\ell_2$ norm between the Successor features of the policies.

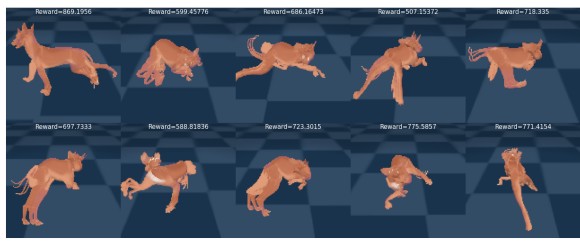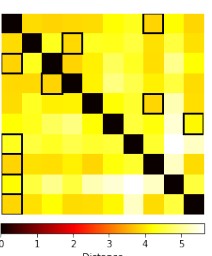

Figure C3: QD in dog.stand. **Left:** 10 policies and corresponding rewards. **Right:** distances in $\ell_2$ norm between the Successor features of the policies.

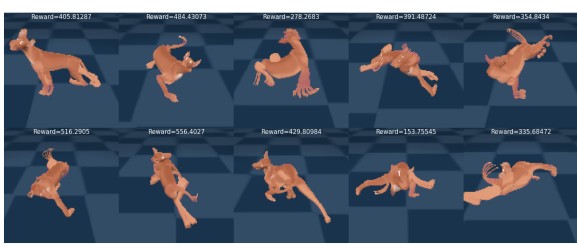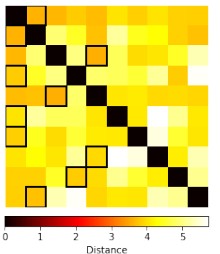

Figure C4: QD in dog.walk. **Left:** 10 policies and corresponding rewards. **Right:** distances in $\ell_2$ norm between the Successor features of the policies.

## C.2 Additional Quality Diversity Results

We now present additional experimental results evaluating the trade-off between quality and diversity using the scatter plots introduced in Fig. 2. y-axis shows the episode return while the diversity score, corresponding to the Hausdorff distance (Eq. (5)), is on the x-axis. The top-right corner of the diagram represents the most diverse and highest quality policies. Each figure presents a sweep over one or two hyperparameters and we use the color and a marker to indicate the values. In all of our scatter plots, we report 95% confidence intervals, corresponding to 5 seeds, which are indicated by the crosses surrounding each point.

**Quality Diversity: walker.walk** In Fig. C5 we show experimental results for DOMiNO in the walker.walk domain. Consistent with Fig. 2 which shows similar results on walker.stand, we show that our constraint mechanism is working as expected across different set sizes and optimality ratios across different tasks.

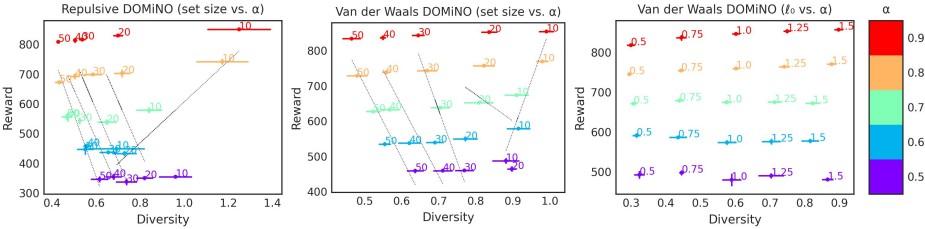

Figure C5: QD Scaling results on walker.walk task. Left: Number of policies vs. optimality ratio in walker.walk with the repulsive reward and (center) with the VDW reward. Right: Optimality ratio vs. VDW target distance $\ell_0$.

**Quality Diversity: SMERL vs DOMiNO** In Fig. C6 we show further experimental results in walker.stand for SMERL in comparison to DOMiNO. When SMERL is appropriately tuned (here for the 10 policies configuration), it can find some solutions along the upper-right QD border; however we find that the best $c_d$ does not transfer to other configurations. The choice of $c_d$ that enables the agent to find a set of 10 diverse policies produces sets without diversity for any other set size.

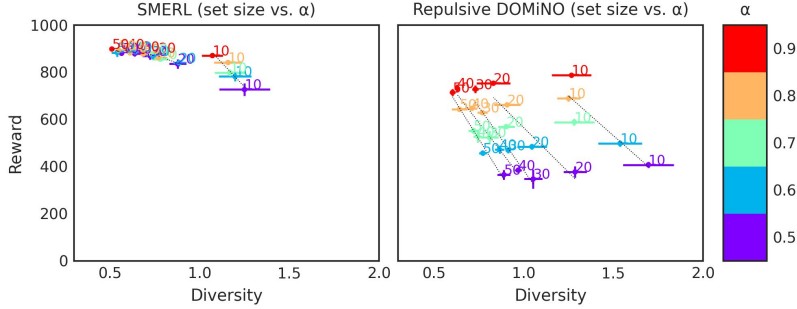

Figure C6: Scaling SMERL (left) vs. DOMiNO (right) on Walker.Stand. Set size is indicated with marker, color corresponds to optimality ratio $\alpha$. The $c_d$ for SMERL is set to 0.5, which was tuned using a set size of 10 policies (see 3, left). This choice does not scale well to any other set size, where regardless of optimality ratios, all policies only optimize for extrinsic reward, at the expense of diversity.

**Discounted State Occupancy** We run the same experiments reported in Fig. 2 with DOMiNO's Lagrangian method and report the results in Fig. C7b. As can be observed, using predicted discounted features does not make any significant difference in performance. Since the mixing time for the DM Control Suite is much shorter than the episode length $T$, the bias in the empirical feature averages is small.

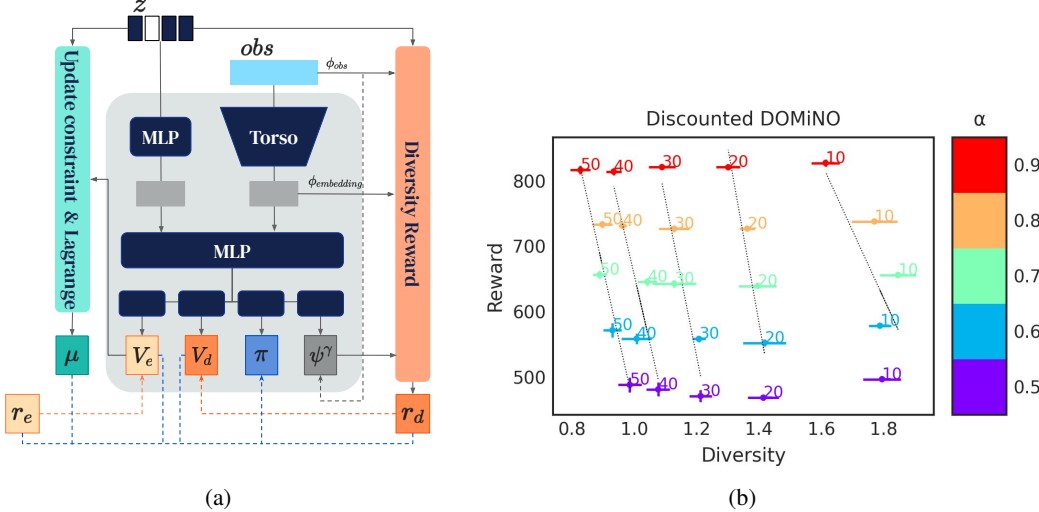

(a)                                                                          (b)

Figure C7: **(a)** DOMiNO with a discounted state occupancy. An additional network head is trained to predict successor features $\psi^\gamma$, which are used instead of the average features $\psi^{avg}$ to compute the diversity reward. The discounted, extrinsic value is used as a constraint instead of the averaged rewards. Dashed lines signify training objectives. **(b)** Number of policies vs. optimality ratio in walker.stand with DOMiNO, consistent with Fig. 2.

**The VDW reward without constraints** Lastly, we study how the VDW reward can help to balance the QD trade off without using the constrained mechanism at all ($r = r_d + r_e$, where $r_d$ is the VDW reward). Fig. C8 presents such study where we compare the VDW contact distance $\ell_0$ and the size of the set. We can see that setting $\ell_0$ to different values indeed gives us this degree of diversity. For example, the purple points correspond to $\ell_0 = 0.25$ and they are all scattered at Diversity values (x-axis) of $0.25$. We can also see that for the same $\ell_0$, increasing the set size decrease performance, and in fact, for large values of $\ell_0$ it is only for small set sizes (of size 10) that it is possible to find sets with that degree of diversity without hurting performance. However this phenomena diminishes for lower values of $\ell_0$. This is expected since if the "volume" of sub optimal policies is fixed, then for lower values of $\ell_0$ it is possible to get more policies inside it.

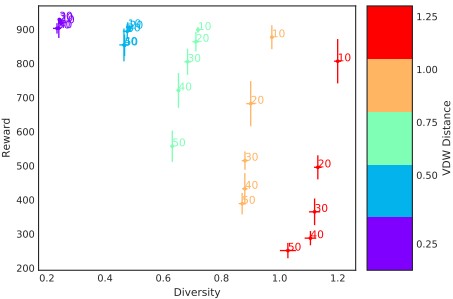

Figure C8: DOMiNO QD results in walker.stand. Set size vs. VDW distance ($\ell_0$) with the VDW reward

**Hyper parameter study of the powers used in the VDW reward Eq. (8)**

As we explained below Eq. (8), the powers in Eq. (7) are chosen to simplify the reward in Eq. (8). The only restriction is that the power of the repulsive force will be lower then of the attractive force. To verify this statement we performed the following hyper parameter study where we changed the attractive power in **3**,4,5 and the repulsive power in **0**,1,2 (bold indicates the values used in the main paper). The results can be found Fig. C9a and Fig. C9b, showing that these hyper parameters have an insignificant effect on diversity and reward. The only impact that we found was on the speed of

convergence of the diversity metric where there are three under performing curves (lower) in Fig. C9b corresponding to the larger attractive force power (2) but still converged at the end to the same level of performance.

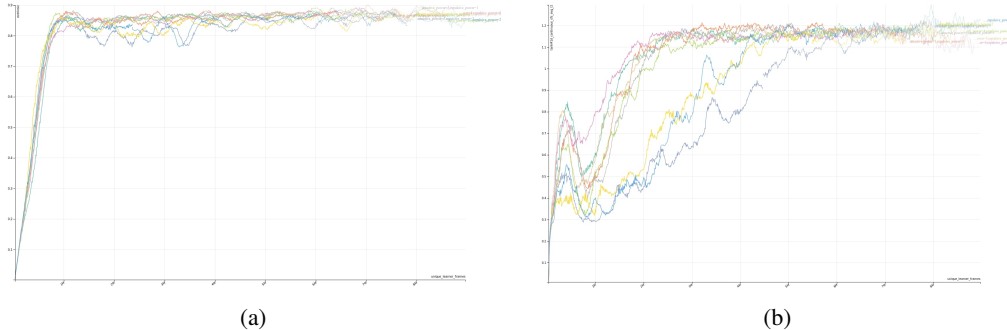

(a)                                                    (b)

Figure C9: Hyper parameter study of the powers used in the VDW reward on the **(a)** Extrinsic reward and **(b)** Diversity.

# D    LIMITATIONS

**Diversity increasing by decreasing** $\alpha$ Inspecting Fig. C5, Fig. C7b and Fig. 2. we can observe that the diversity score increases for lower optimality ratios. Recall that the optimality ratio $\alpha$ specifies a feasibility region in the state-occupancy space (the set of all $\alpha$-optimal policies). Thus, the size of this space increases as $\alpha$ decreases, and we observe more diverse sets for smaller values of $\alpha$. This intuition was correct in most of our experiments, but not always (e.g., Fig. C5).

One possible explanation is that the Lagrange multipliers solution is seeking for the lowest value of $\lambda$ that satisfies the constraint (so that we can get more diversity), i.e., it finds solutions that satisfy the constraint almost with equality: $v_e^i \sim \alpha v_e^*$ (instead of $v_e^i > \alpha v_e^*$). The size of the level sets ($v_e^i = \alpha v_e^*$) do not necessarily increase with lower values of $\alpha$ (while the feasibility sets $v_e^i \geq \alpha v_e^*$ do). Another explanation is that in walker.walk (Fig. C5) it might be easier to find diverse walking (e.g., $\alpha = 0.9$) than diverse "half walking" (e.g., $\alpha = 0.5$). This might be explained by "half walking" being less stable (it is harder to find diverse modes for it).

**Features** Another possible limitation of our approach is that diversity is defined via the environment features. We partially addressed this concern in Table 1 where we showed that it is possible to learn diverse high quality policies with our approach using the embedding of a NN as features. In future work we plan to scale our approach to higher dimensional domains and study which auxiliary losses should be added to learn good representations for diversity.

# E    ADDITIONAL K-SHOT EXPERIMENTS

## E.1    GENERAL COMMENTS

We note that while we tried to recreate a similar condition to (Kumar et al., 2020), the tasks are not directly comparable due to significant differences in the simulators that have been used as well as the termination conditions in the base task.

For CI, we use boosted CI with nested sampling as implemented in the bootstrap function here, which reflects the amount of training seeds and the amount of evaluation seeds per training seed.

## E.2    CONTROL SUITE

Next, we report additional results for K-shot adaptation in the control suite. In Fig. E10 we report the absolute values achieved by each method obtained (in the exact same setup as in Fig. 4). That is, we report $r_{\mathrm{method}}$ for each method (instead of $r_{\mathrm{method}}/r_{\mathrm{baseline}}$ as in Fig. 4). Additionally, we report $r_{\mathrm{baseline}}$, which is the "Single policy baseline" (blue) in Fig. E10. Inspecting Fig. E10, we can see that all the methods deteriorate in performance as the magnitude of the perturbation increases. However, the performance of DOMiNO (orange) deteriorates slower than that of the other methods. We can also see that the performance of the no diversity baseline is similar when it learns 10 policies (red) and a single policy (blue), which indicates that when the algorithm maximize only the extrinsic reward, it finds the same policy again and again with each of the 10 policies.

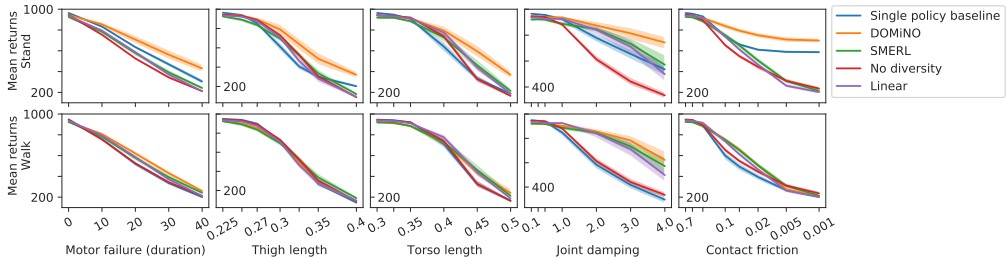

Figure E10: K-shot adaptation in Control Suite, similar to Figure 4, but reporting absolute rather than relative returns.

Next, we inspect a wider range of hyper parameters for the SMERL and linear combination methods. Concretely, for DOMiNO and SMERL $\alpha = 0.9$, for SMERL $c_d \in [0.5, 1, 2, 5]$ and for linear

combination $c_e \in [0.1, 0.2, 0.3, 0.4, 0.5, 0.6, 0.7, 0.9]$ and all methods are trained with 10 policies. These values correspond to the values that we considered in Fig. 3.

Inspecting Fig. E11 we can see that the best methods overall are DOMiNO and SMERL (with $c_d = 1$). We can also see that DOMiNO and SMERL consistently outperform the linear combination baseline for many hyper parameter values. This is consistent with our results in Fig. 3 which suggest that the linear combination method tends to be either diverse or high performing and fails to capture a good balance in between. Lastly, it is reasonable that SMERL and DOMiNO perform similar since they are both valid solutions to the same CMDP. However, SMERL comes an with additional hyper parameter $c_d$ that may be tricky to tune in some situations. For example, trying to tune $c_d$ based on the results in the vanilla domain (picking the upper-right most point in Fig. 3) led us to choose $c_d = 0.5$ for SMERL, instead of 1. The Lagrange multipliers formulation in DOMiNO does not have this challenge as it does not have an extra hyper parameter.

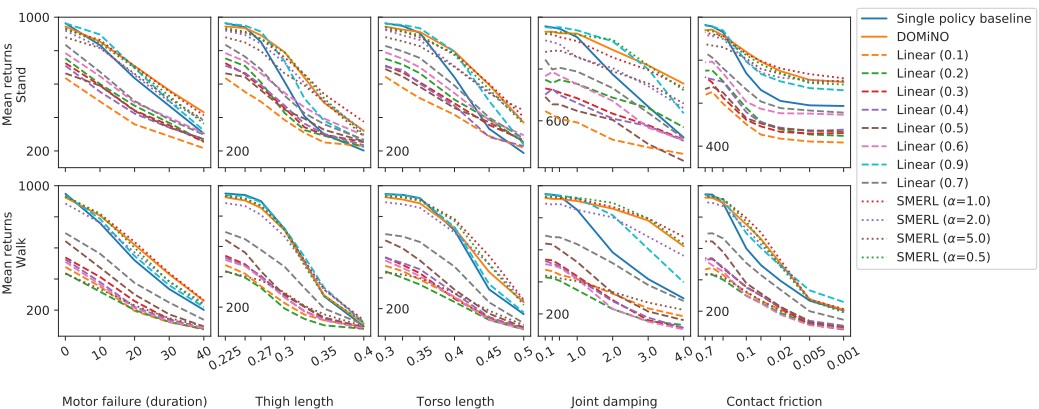

Figure E11: K-shot in Control Suite, similar to Figure 4, but reporting a wider range of hyper parameters for SMERL and linear combination.

### E.3 VARIABLE K.

Next, in Fig. E12 we performed an ablation study on the impact that the total number of trajectories $K$ has on selecting the best policy in the set. Since $K$ is the total number of trajectories, it is not always perfectly divided by the number of policies (chosen to be 10 for this experiment). Thus, we randomly distribute the modulo between the policies. Fig. E12 suggests that using more trajectories clearly helps, but it also suggests that performance saturates early on and that it is possible to use a much smaller $K$ (around 30) and still outperform the "single extrinsic policy baseline".

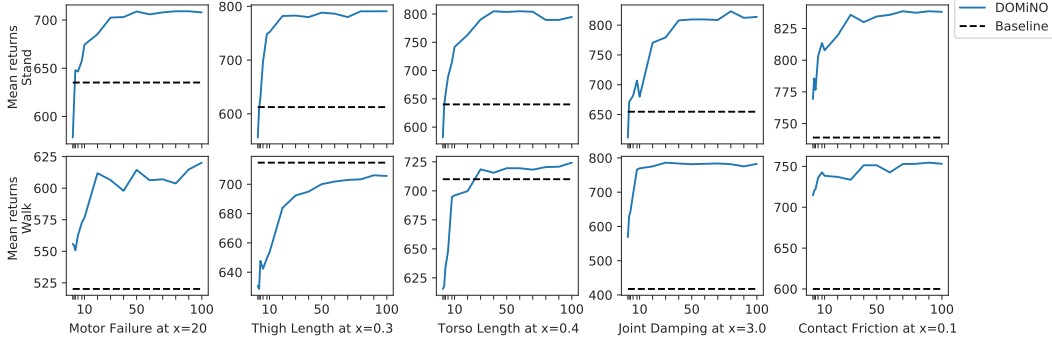

Figure E12: K-shot adaptation ablation. K, the number of total trajectories (x-axis) vs the performance of the best policy in the set (y-axis). Baseline corresponds to the performance of the single extrinsic only policy.

### E.4 BIPEDALWALKER

For the **BipedalWalker** environment, we either perturb the morphology or the terrain. To perturb the morphology, we follow (Ha, 2018) and specify a set of re-scaling factors. Specifically, each leg is made up of two rectangles, with pre-defined width and height parameters: $\text{leg}_1 = ((w_1^1, h_1^1), (w_1^2, h_1^2))$, $\text{leg}_2 = ((w_2^1, h_2^1), (w_2^2, h_2^2))$. To generate a perturbed morphology, we define a scaling range $[0, \eta]$ withing which we uniformly sample scaling factors $\ell_i^j, \nu_i^j \sim [-\eta, \eta]$, for $i = 1, 2$ and $j = 1, 2$. A perturbed environment is defined by re-scaling the default parameters: $\widetilde{\text{leg}}_1 = (((1 + \ell_1^1)w_1^1, (1 + \nu_1^1)h_1^1), ((1+\ell_1^2)w_1^2, (1+\nu_1^2)h_1^2)))$, and $\widetilde{\text{leg}}_2 = (((1+\ell_2^1)w_1^1, (1+\nu_2^1)h_1^1), ((1+\ell_2^2)w_1^2, (1+\nu_2^2)h_1^2)))$. The values for this perturbations can be found in Table 4.

| Perturbation type | Perturbation scale parameter values ($\eta$) |
|---|---|
| Morphology | 0., 0.10, 0.15, 0.20, 0.25, 0.30, 0.35 |
| Stumps (height, width) | 0., 0.1, 0.2, 0.3, 0.4, 0.5, 0.6, 0.7, 0.8, 0.9, 1. |
| Pits (width) | 0., 0.1, 0.2, 0.3, 0.4, 0.5, 0.6, 0.7, 0.8, 0.9, 1. |
| Stairs (height, width) | 0., 0.1, 0.2, 0.3, 0.4, 0.5, 0.6, 0.7, 0.8, 0.9, 1. |

Table 4: Bipedal perturbation scale values

For terrain changes, we selectively enable one of three available obstacles available in the OpenAI Gym implementation: stumps, pits, or stairs. For each obstacle, we specify a perturbation interval $[0, \eta]$. This interval determines the upper bounds on the obstacles height and width when the environment generates terrain for an episode. For details see the "Hardcore" implementation of the BiPedal environment. Note that for stairs, we fixed the upper bound on the number of steps the environment can generate in one go to 5.

To evaluate adaptation, we first train 10 agents independently on the "BiPedalwalker-v3" environment, which only uses a flat terrain. To evaluate the trained agents we sample random perturbations of the environment. Specifically, for each type of perturbation (morphology, pits, stumps, stairs) and for each value of the scale parameter $\eta$, we randomly sample 30 perturbations. We then run each option for 40 episodes; adaptation takes the form of using the first 10 episodes to estimate the option with highest episode return, which is then used for evaluation on the remaining 30 episodes.

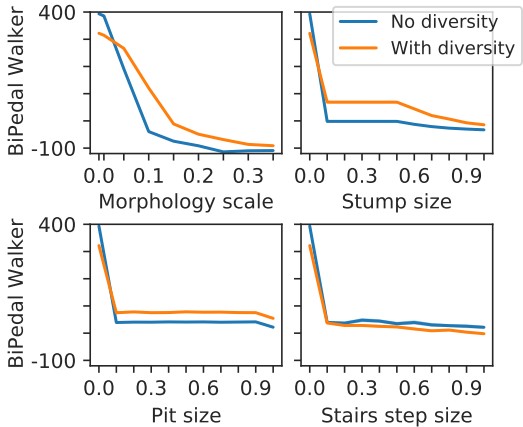

Figure E13: K-shot adaptation in BiPedal walker

Fig. F15 shows that, while performance degrades as the morphology is deformed, DOMiNO exhibits greater adaptability as evidenced by less severe degradation of performance. In terms of morphology, we find a gradual decline in performance as we increase the degree of deformation. Similar to the Control Suite, diversity is beneficial and helps the agent adapt while not being impervious to these changes. In terms of terrain perturbations, these have a more abrupt impact on the agent's performance. While diversity does not prevent a significant drop in performance, it is still beneficial when adapting to stumps and pits and does not negatively impact performance in the case of stairs.

## F    ADDITIONAL FIGURES FOR THE REBUTTAL

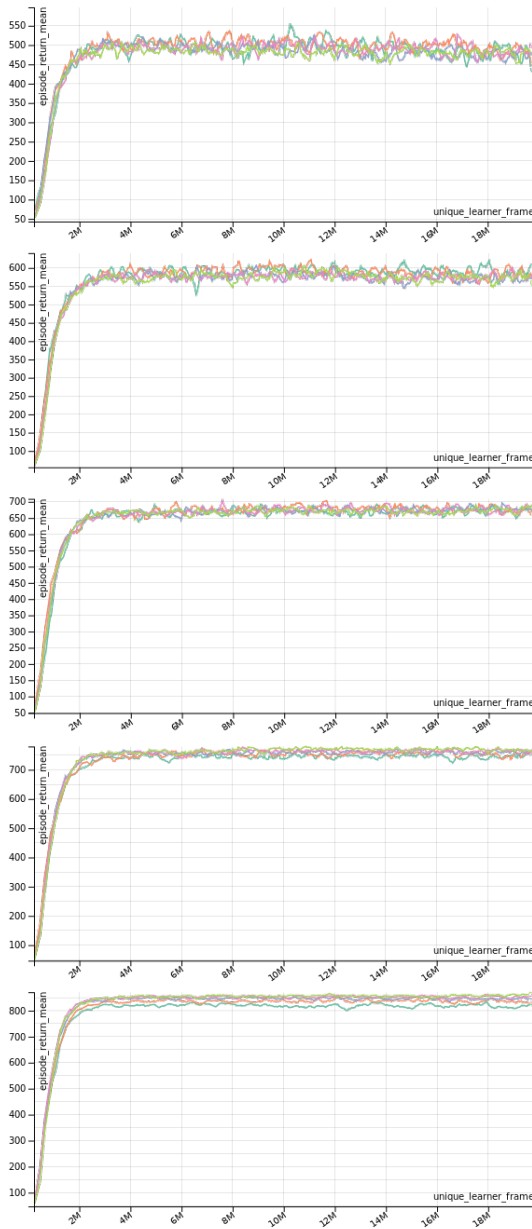

Figure F14: Reward learning curves (episode reward vs number of environment steps) corresponding to the results in Figure 2, center. Sub figures correspond to different optimally ratio (0.5 to 0.9 up to bottom), colors correspond to values of $\ell_0$.

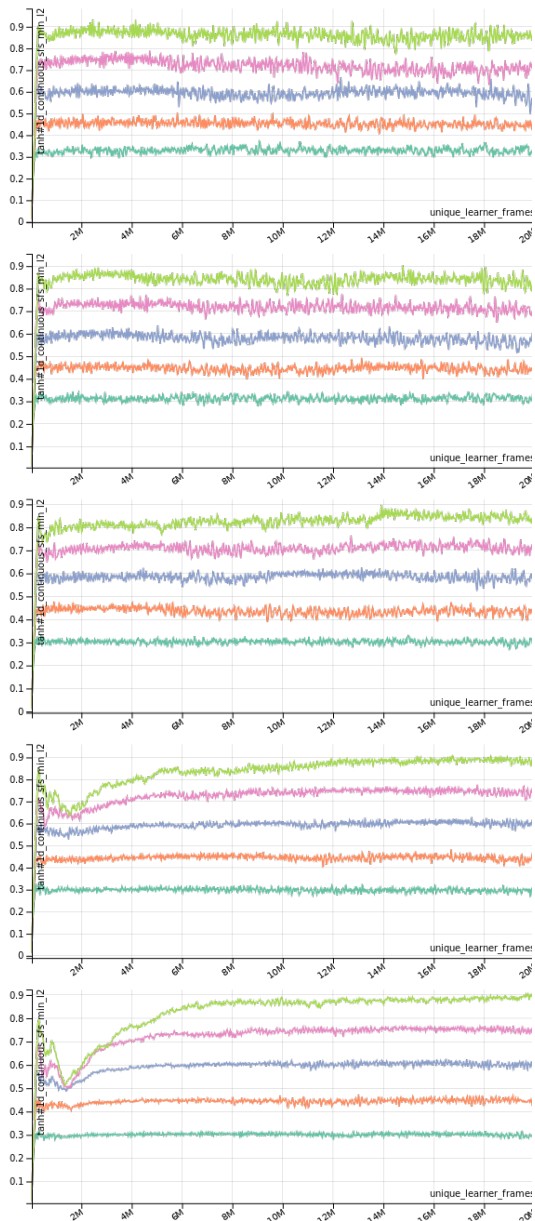

Figure F15: Diversity learning curves (episode reward vs number of environment steps) corresponding to the results in Figure 2, center. Sub figures correspond to different optimally ratio (0.5 to 0.9 up to bottom), colors correspond to values of $\ell_0$.

