# OpenReview forum: "Discovering Policies with DOMiNO: Diversity Optimization Maintaining Near Optimality"
_ICLR.cc/2023/Conference — ICLR 2023 poster_

### Official Review · Reviewer_vof2 · 2022-10-23

**Confidence:** 4
**Correctness:** 3
**Technical Novelty And Significance:** 3
**Empirical Novelty And Significance:** 3
**Recommendation:** 6

**Clarity, Quality, Novelty And Reproducibility:**

To the best of my knowledge, the work proposes a novel formulation of diverse skills discovery through a direct distance between state-action occupancies induced by the policy, which is interesting. Thus, I think originality is a strength of this paper. The motivations for this work are well-reported, but the underlying theoretical ground and the proposed methodology could be described with more details and clarity.

**Strength And Weaknesses:**

*After discussion*

Having read the authors' response and other reviews, I am changing my score from 5 to 6.

----

*Strengths*
- (Novelty and relevance) The paper introduces an interesting objective that measures the diversity between the policies through a direct distance in the space of the state-action distributions. While previous works have tackled diverse skills discovery, this direct formulation is, to the best of my knowledge, novel;
- (Experimental analysis) The proposed approach is evaluated in challenging domains, and the experiments' section includes nice visualizations and interesting findings.

*Weaknesses*
- (Theoretical ground) The theoretical ground for the proposed framework is weak. Especially, the paper proposes to maximize non-concave objectives, and convergence guarantees for the approach are questionable;
- (Algorithm) The paper dedicates little space to the presentation of the algorithm, which should be a relevant contribution per se given the novelty of the problem, and the corresponding technical challenges (e.g., how the objective function can be estimated from samples);
- (Comparison with previous works) Whereas the authors made clear that the experimental analysis is not intended to demonstrate the superiority against any baseline, previous works addressed both reward robustness as well as diverse skills discovery, and even an indirect comparison could have been telling.

*Detailed Comments*

Optimization Problem

(C1, Major) The discovery of near-optimal policies is formulated through a maximization problem of a non-concave function, e.g., the L2-distance between the expected state-action features. While the objective is at least convex, and thus the gradient ascent procedure may work fine within the space, it is unclear what happens at the boundaries. This problem has been pointed out before, such as in (Eysenbach et al., 2021), which is mentioned in the paper. Especially, I am wondering whether the optimization problem proposed in the paper is tractable even when the underlying MDP is fully known. Can the authors assess the computational complexity of the problem?

(C2) Also related to the previous comment: The optimal set of policies is attained by deterministic Markovian policies in this setting? To my intuition, if I am just maximizing the L2-distance between the occupancies of two policies, I should converge to deterministic policies. However, it is not obvious to me what happens with more than two policies and the value constraint.

(C3, Major) Even ignoring the potential intractability of the problem, I am wondering what kind of game-theoretic guarantees do we have if all the policies are optimized simultaneously via gradient ascent. Have the authors some ground to say that the optimization would converge to some notion of equilibrium?

(C4) The paper proposes two different objective formulation, either the L2-distance or Van Der Waals force. Can the authors better motivate why these two alternative have been chosen over other distances/divergences? Do we necessarily need a proper distance for the objective function? KL-divergence, which is not symmetric, could work as well?

(C5, Minor) The problem is formulated through a maximization of the diversity objective under the value constraint. Do the authors considered also an alternative formulation though a set of problems in which the objective is the value of a policy under a diversity constraint? Would this formulation change the nature of the problem to make it (possibly) tractable?

Methodology

(C6) The methodology is not clearly explained in the main text, as it is mostly relegated to the Figure 1a, in which the architecture is presented, and a few lines at the end of Sec. 3.3 and the Agent paragraph in Sec. 4. Can the authors describe the methodology in details, and what kind of peculiar challenges it has to address?

(C7) Especially, I am wondering how problematic it can be to estimate the objective function in practice. Estimating a distance/divergence through density estimations is known to be cumbersome in continuous/high-dimensional domains. Do the authors validated their estimators empirically?

(C8, Minor) Previous works in imitation learning (e.g., Schroeker and Isbell, State aware imitation learning, 2017; Schroeker et al., Generative predecessor models for sample-efficient imitation learning, 2018) have considered the gradient of the L2 distance between state-action occupancies. Clearly, they follow a gradient descent procedure instead of the gradient ascent proposed in this paper to maximize diversity, but the gradient formula could still have some common traits.

Experiments

(C9, Major) Whereas the authors explained that they do not intend to compare their approach to any baseline, it is quite hard to actually evaluate the results, as we do not know how far the obtained policies are from the global optimum. Especially,
- Can the authors show the value of the objective during training, to understand whether the algorithm is actually improving the value of the objective with a monotonic trend?
- Can the authors show that the algorithm is converging, or do they just stop the training at some point while the policies would continuously change?

Related Works

(C10) While there is not a clear baseline for the proposed approach, as the problem formulation is novel to my knowledge, there exist previous methods that addresses similar problems. Especially,
- Maximum entropy regularization for RL is known to provide policies that are reward-robust (see Husain et al., Regularized policies are reward robust, 2021);
- The proposed objective with a void value constraint is similar in nature to the problem of (unsupervised) diverse skills discovery (e.g., Eysenbach et al., 2019).

(C11, Minor) Another recent work (Mutti et al., Reward-free policy space compression for reinforcement learning, 2022) comes up with an optimization problem that have some similarities with the one proposed here, although the underlying motivation is different.

**Summary Of The Paper:**

This paper addresses the problem of learning a set of diverse policies that are simultaneously nearly-optimal for a given reward function. In their formulation, the diverse near-optimal policies are learned by maximizing an objective function related to diversity, while near-optimality is ensured through a linear constraint on the value of each policy. Crucially, the diversity between the policies is measured through a distance (either L2 or Van Der Waals force) of the expected state-action features they induce. The paper proposes to optimize the introduced problem through a Lagrangian relaxation that can be seen as a three-player game. Finally, the paper provides an experimental analysis in the DeepMind Control Suite to show that
- the proposed approach leads to the discovery of diverse nearly-optimal behaviors, where near-optimality and diversity can be traded-off in various ways;
- the proposed Lagrangian approach dominates a simple scalarization of the diversity and near-optimality objectives;
- the set of diverse nearly-optimal policies allows for efficient k-shot adaptation to reward perturbations.

**Summary Of The Review:**

The idea proposed in this paper is interesting and original, and it has interesting connections with several previous works in the literature. On the one hand, it can be seen as a formalization of skills discovery methods when the reward constraint is set to void, whereas it is related to constrained convex RL otherwise, and to multi-objective RL if we scalarize the combination of the objective and constraint. It has also technical similarities with apprenticeship learning and state-based imitation learning.

However, the provided theoretical ground for the approach is somewhat weak: It is not clear if the proposed optimization problem is tractable, and what kind of guarantees can we have optimizing it. The algorithmic contribution is also relegated to tiny bits of the paper, while it should be a main contribution to my understanding.

Overall, my current evaluation for this work is borderline, but I am open to raise my score if the authors could address my comments above.

---

> ### Author Response · Authors · 2022-11-15
> **Official response**
>
> We woud like to thank the reviewer for finding our work original, the formulation to be interesting and the visualizations to be nice.
>
> C1-C3
> As the reviewer mentioned, the problem of maximizing diversity is not convex even in the tabular RL setting. While there has been some progress in non convex-concave saddle point optimization lately, analyzing this setup theoretically was not the focus of our work, and to the best of our knowledge, has not been addressed before in the context of diverse skill discovery. Our answers are therefore based on intuition and empirical evidence.
>
> C1: in light of the above, these properties will not change even if we consider the tabular or known MDP scenario, as the problem will remain non convex. The reviewer is correct that maximizing a convex function via gradient ascent should in principle take us to the boundary of the set. Once the algorithm reaches the boundary, it might move along the boundary till it finds a good equilibrium between the policies. However, for different choices of hyper parameters it is also possible that the algorithm will oscillate away from the boundary and even diverge given that the problem is not convex. In practice, we did not observe these problems.
>
> C2: regarding the determinism of the solutions. If we consider the convex setting, then there are no guarantees that the algorithms used in this paper for minimizing a convex objective will return a deterministic policy and in general, it should return a stochastic policy. In addition, there are some convex objectives for which it is known that the optimal policy is stochastic (e.g., max entropy RL). It is only in the standard (linear) RL setup, that it is known that there is always a deterministic optimal policy and that there are algorithms for finding it. Now, if we think about the state-action occupancy polytope, and only two policies, then we agree with the reviewer's intuition that the two furthermost points on this polytope should be two nodes (deterministic policies).
>
> However, this is not true in general. For a negative example consider a quadrilateral polytope with kite geometry. Let's denote the nodes (deterministic policies) A,B,C,D such that the long diagonal is AB and the short diagonal is CD, and the intersection of the diagonals O. Recall that since its a kite we have that AC=AD and CB=BD. Lastly, let's take the limit where AO>>OB, in which case AC=AD~CD/2. In this case the diversity of the set of deterministic policies A,B,C,D (the minimum distance between two points) is CD/2. Now consider the following set composed of the points A,B and two stochastic policies C’,D’ where C’ is an intermediate point along the CB line and D’ is an intermediate point along the DB line. It is easy to see that the distances AC’ and AD’ are larger than CD/2. In addition, as long as C’ is close enough to C the distance C’D’ is close to CD and larger than CD/2. Thus the minimum distance between two points is larger for the set ABC’D’ (which includes two stochastic policies) than the set ABCD which includes only deterministic ones.
>
> C3: considering the case that the objective is jointly convex in the state occupancies of all the policies in the set, it should be possible to guarantee that all the policies converge on average. Here, by average, we mean that if we consider the policies produced by the algorithm at different iterations, and choose one of them at random (the mixed policy) then the mixed policies converge to an equilibrium. However, it is not clear if the same algorithms can guarantee last iterate convergence (the convergence of the last policy returned by the algorithm) and might oscillate in practice.
>
> C4: in this paper we focused on diversity objectives that can be optimized with Successor Features via the FTL cost player. Other distance metrics like the KL can also be chosen, but are not tractable when used with algorithms proposed in this work. For example, differentiating the KL directly to produce an intrinsic reward involves a softmax over states. That said, it is possible to optimize the KL with other methods like in DIAYN and VIC using variational inequalities and introducing a discriminator. Also see our reply to C10 below.
>
> C5: this is an interesting formulation that has been studied in related work but we did not study it in this work. Our intuition is that it won’t make the problem tractable: instead of maximizing a non convex objective under linear constraints it will result in standard RL (which is fine) under a non convex constraint (which is intractable).

---

> > ### Author Response · Authors · 2022-11-15
> > **Official response cont.**
> >
> > C6: thank you for pointing this out. We have added many implementation and algorithmic details in the supplementary material but could not fit everything in the main paper due to the page limit. Given that an extra page will be allowed for the camera ready copy, we will make sure to move some of these to the main paper. We hope that this is addressing your question, but if you have any specific questions please let us know.
> >
> > C7: in practice, we estimate the successor features of each policy either with a learned value head or with moving averages of features. See section C2 for more details and comparisons. Overall we find both methods to perform similar and yield nearly optimal diverse policies, although, it is possible that there are some hidden challenges that we did not notice. Regarding empirical verification, we measure the diversity between the SFs of the policies in the set and report it in most of our experiments, see Figure 2 and similar figures.
> >
> > C8: It is true that diversity and imitation learning objectives have great similarities and can be seen as opposites of each other. Imitation learning aims to find a policy that minimizes the difference in behavior to another while diversity tries to find one that maximizes said difference. This similarity can be seen in prior methods: where GAIL trains a discriminator to distinguish between expert and student, DIYAN trains a discriminator to distinguish between diverse policies. As for the specific methods cited above, these aim to estimate the gradient of the KL-divergence (not L2) by training a model on data from one policy (the student) and evaluating it on data from another (the expert). A discriminator-based approach on the other hand, would be trained on data from both policies. This asymmetry is advantageous for sample-complexity when data from one policy is more expensive to obtain than from another as is usually the case in imitation-learning, but not in our setup. The disadvantage is that the learned model will be inaccurate when the two policies are too far apart; something that can be worked around in imitation learning but not for diversity.
> >
> > C9
> > While the reviewer is correct that in some cases, it is hard to tell how far are our policies from the optimal point, we would like to point out the reviewer to a few experiments with the constraint mechanism and the VDW reward do suggest that our algorithm indeed finds the solution to the optimization problem. For example, in Figure 2 (right) it is possible to see that for each $\ell_0$ that we specify as a hyper parameter for the VDW reward, the diversity of the discovered set is indeed very close to $\ell_o$. Similarly for constraint satisfaction, the reviewer can examining the reward of the individual policies (at the top of each sub figure in figures C1-C4) and see that they are all very close to satisfy the constraint, ie, they achieve reward of around 0.9 of the “only extrinsic reward” maximizing policy that is always present first in the top left corner.
> >
> > For learning curves, we uploaded the learning curves corresponding to the experiment in Figure 2 (center) in the end of the supplementary material. There is a sweep over the VDW coefficient $\ell_0$ (different colors) and the optimality ratio (sub figures) and the exact values are specified in the paper. The learning curves are averaged over five seeds. The reviewer can see that on average the method is pretty stable.

---

> > > ### Comment · Reviewer_vof2 · 2022-11-18
> > > **After Response**
> > >
> > > I want to thank the authors for their detailed replies to my questions and comments. In the answer to C4, authors refers to the reply to C10, which I cannot find in the subsequent text box. Did they perhaps forget to add a portion of their response?
> > >
> > > *Figures F14, F15*
> > >
> > > The additional Figure F14 is clear, what does F15 show instead? The caption reports episode rewards vs number of environment steps, but I guess it is referring to diversity between policies (which policies exactly?)
> > >
> > > How are the policies initialized in this experiment?

---

> > > > ### Author Response · Authors · 2022-11-18
> > > > **Reply**
> > > >
> > > > Thank you for following up with us and sorry about the confusion.
> > > >
> > > > In the answer to C4 we were referring to C8. Our argument is that the KL between state occupancies can be maximised with other algorithms (DIAYN, VIC, the references by Schroeker et al.) but these algorithms are different algorithmically from the methods we proposed in this paper. Our focus in this paper is on on metrics that can optimised using the gradient of the metric as intrinsic reward and estimate it this intrinsic reward with Successor features.
> > > >
> > > > Regarding C10, thanks for referring us to these related work. The reviewer is correct that there has been a significant amount of work on diverse skill discovery using the KL distance, which we cited in the main paper. Our work demonstrates that it is also possible to use other metrics for diverse skill discovery and proposes an algorithmic framework for this, which we believe is a novel contribution.
> > > >
> > > > Regarding the learning curves, Figure F15 shows the diversity of the set on the Y-axis, measured via the average Hausdorf distance. That is, the L2 distance between the successor features of a policy, and the policy that is the closest to it in the set (Equation 5). The distance is reported on average over the set. The different colors correspond to different VDW coefficients $\ell_0 \in $ and notably, reach different Hausdorf distances as a result.
> > > >
> > > > The policies are represented as a latent conditioned architecture (Figure 1A), and the parameters of the network are initialised with default values. Concretely, for each linear layer we set stddev = 1. / np.sqrt(self.input_size) and use w_init = initializers.TruncatedNormal(stddev=stddev).

---

> > > > > ### Comment · Reviewer_vof2 · 2022-11-18
> > > > > **Follow-up**
> > > > >
> > > > > Thank you for the clarifications. With the new pieces of information I am now feeling more confident in my evaluation of the paper. Especially, I am reassured by the additional experiments showing that the learning process is mostly stable, and that the algorithm makes significant improvement w.r.t. the initialization. The remarks about convergence of the average mixture policies and stochasticity of the optimal policies are also interesting: I suggest the authors to further develop on them and to include their results in a final version of the paper.
> > > > >
> > > > > Overall, I think this paper is valuable, and I am updating my score accordingly.
> > > > > I still believe there is room for improvement in terms of:
> > > > > - theoretical characterization of the problem,
> > > > > - including a clearer description of the algorithmic procedure and its guarantees,
> > > > > - a deeper empirical analysis of the k-shot adaptation setting,
> > > > > - an experimental (qualitative) comparison with related methods (such as skills discovery).

---

### Official Review · Reviewer_oggL · 2022-10-23

**Confidence:** 4
**Correctness:** 2
**Technical Novelty And Significance:** 3
**Empirical Novelty And Significance:** 3
**Recommendation:** 6

**Clarity, Quality, Novelty And Reproducibility:**

Regarding clarity, although I enjoy reading the technical content to learn all the detailed derivations, I do feel the entire paper looks unnecessarily complicated, particularly for the general audience. The paper reads like a technical tutorial, which makes a clear attempt to explain every single step to a reader but doesn't really look like a conference paper. Unfortunately, I don't really have very concise constructive feedback, so I just express my feelings here.

The technical quality and novelty are good. I also believe the results are reproducible.

**Strength And Weaknesses:**

## Strength
I do enjoy reading the paper from a technical perspective and feel really interested to see the conclusion that the derived algorithm can be simply built upon the gradient of diversity measure, which is novel in my opinion.

The experiments are sufficient and solid from my perspective. The section contains sufficient ablation studies, emergent behaviors as well as the full suites of experiments in the original SMERL paper.

## Weakness
My major concerns are about the presentation of this paper. Some of the details are listed below.

1. The title is definitely a bad one. I do understand the space is pretty limited, but I still think the authors could have done a better job to come up with a more informative title, i.e., at least including the terms "diverse" and "nearly optimal". It is really a misleading title, although I do know this will be addressed in the camera-ready version.

2. I have to express my disagreement with the statement "we do not explicitly compare it with other work nor argue that one works better than the other." This is really an awkward statement in a paper, particularly knowing that it is even presented twice! I could understand that there could be some reason, but I do believe there are better ways to position this work in the literature. Note that you do compare with SMERL and outperform it! It is completely okay to say this work is better than SMERL, isn't it? Also, wouldn't it be natural to include a few SMERL follow-ups as additional baselines for improved soundness? The only tricky part of baselines that I could imagine would be those diversity methods on a swapped objective, i.e., maximizing the extrinsic reward with a diversity constraint. This line of research is related but, in fact, parallel to this work and will result in a completely different solution set. More interestingly, the algorithmic framework here will no longer hold if the diversity and reward objectives are swapped. So, I would suggest the authors spend some texts discussing the difference between the two lines of work (i.e., SMERL v.s. diversity MARL) in the main paper for better paper positioning rather than repeatedly leaving such an awkward execuse.

3. **A technical flaw in VDW objective**. I want to point out the derivation holds under the assumption of convex MDP. However, in equation (7), the diversity metric induced by the Van der Waals force isn't really convex over the entire domain --- it is only convex in a sub-space. Although in practice, it seems that this doesn't really matter much, I do think this part should be fixed or at least be discussed further.

4. **novelty and positioning**: A large body of the content, particularly the algorithmic framework, directly follows [1]. In the current draft, the whole related work section is deferred to the appendix without a careful discussion of the relationship between this work and [1]. To me, it looks like the authors adopt the framework from [1] and apply it to the SMERL setting. I do think this should be explicitly presented in the introduction section.

[1] Reward is enough for convex MDPs, Tom Zahavy, Brendan O'Donoghue, Guillaume Desjardins, Satinder Singh, NeurIPS 2021.

**Summary Of The Paper:**

This paper considers the problem of jointly learning a diverse set of (nearly) optimal policies within a single RL environment. The paper follows the SMERL paper and adopts a novel mathematical framework based on convex MDP and Fenchel duality. The derived algorithm shows that we can simply solve the constrained optimization by designing an intrinsic reward based on the successor feature of the learning policy and the gradient of the diversity measure.

**Summary Of The Review:**

This is a technically interesting paper with solid experiments, although there still exist some flaws. Assuming the paper will be improved by the authors, my current perspective is leaning towards acceptance since I did learn something from this paper.

---

> ### Author Response · Authors · 2022-11-15
> **Official response**
>
> We would like to thank the reviewer for finding our problem formulation and solution novel and the experiments to be solid. We are also happy to hear that you enjoyed reading the paper.
>
> Positioning and convexity (weakness 3 and 4). As the reviewer pointed out, solving a convex MDP by maximizing the gradient of the objective as an intrinsic reward is present in previous work. That said, diversity maximization (for example, via the Hausdorff or the VDW objectives) is not a convex RL problem (the opposite problem of minimizing diversity has a convex formulation). Thus, the empirical contributions of this work are the study of the application of convex RL methods to a non convex RL problem. Our empirical findings suggest solid evidence that this is indeed a feasible approach which might open the community of diversity in RL to a new class of algorithms. For example, Figure 2 (right) suggests that DOMiNO indeed solves the problem it is designed to solve: it satisfies the constraint and achieves the $\ell_0$ diversity specified in the VDW reward. This is perhaps not surprising: the Deep Learning literature sometimes builds on optimization algorithms that were developed to solve convex problems to great success. Nevertheless we believe that our paper presents an important demonstration of this principle that is important to the study of diversity in RL. We will make sure to highlight this discussion in the paper as well.
>
> We are sorry to hear that the reviewer finds some of our claims to be “awkward”, we will make sure to fix this. Our main goal in this paper was to show that diversity can be optimized efficiently and balanced with quality, via convex RL methods. We did not feel that a comparison between our method and some other baselines would serve this goal and did not want to make such claims even if true, simply because this was not our focus. That said, reading the reviews we understand that the reviewers disagree with us on this point and we now know better the reasons for that. We will make sure to clarify this in the main paper and remove the currently “awkward claims''. We will also make sure to include a discussion on diversity MARL and SMERL follow ups. We summarized all the related work that we are aware of in Section A in the supplementary material, but if the reviewer is aware of some other work that we missed please point us to them.
>
> Lastly, we will make sure to change the title to be more informative.

---

> > ### Comment · Reviewer_oggL · 2022-11-16
> > **Regarding Convex v.s. Non-Convex**
> >
> > Thanks for explaining this concept of adapting a convex MDP algorithm to the non-convex setting. I think this is a fair argument, and I strongly encourage the authors to include this discussion in the main paper to clarify this perspective to avoid possible confusion.

---

> > > ### Author Response · Authors · 2022-11-16
> > > **Updated version with discussion on non convexity, following the rebuttal**
> > >
> > > Thank you, we have added two paragraphs to discuss and clarify this perspective in the conclusion paragrpah. You can find the new paragraphs in blue color.

---

### Official Review · Reviewer_9fff · 2022-10-25

**Confidence:** 3
**Clarity, Quality, Novelty And Reproducibility:** As far as I know this approach to gen…
**Correctness:** 4
**Technical Novelty And Significance:** 3
**Empirical Novelty And Significance:** 3
**Recommendation:** 6

**Strength And Weaknesses:**

Strengths:
* Reformulating the problem of promoting diversity through Fenchel duality is novel and interesting
* Using this reformulation to recast the problem as a multi-agent RL problem allows flexibility of the method as algorithms change

Weaknesses:
* It is difficult to understand the quality of the diversity metrics without more extensive qualitative evaluation.  One could expect a state-occupancy based diversity measure could be degenerate in the same way the noisy-TV problem effects surprise based exploration.  If put into a best buy full of TVs, it seems like each policy could learn to look at a different TV, ignoring many non-TV looking behaviors.
* Sometimes multi-agent and single-agent problem formulations are confused, for instance in Section 3.3 the reduction is to a 3-player game, so it is not a reduction to an MDP, but to a Markov Game.


**Summary Of The Paper:**

This paper proposed a method for training diverse populations fo agents, using any RL algorithm, through a carefully constructed adaptive reward function.

**Summary Of The Review:**

I'm recommending a weak accept for this paper due to the novel framework for optimizing diversity based methods, which appears to be practical for optimizing the objectives it was tested on.

---

> ### Author Response · Authors · 2022-11-15
> **Official response**
>
> We would like to thank the reviewer for finding the problem of promoting diversity through Fenchel duality as novel and interesting and to allow future and flexible research.
>
> For qualitative evaluation of the quality diverse policies we would like to refer the reviewer to the videos we submitted in the supplementary material and to the “motion figures” (C1-C4 in the supplementary). These show that the discovered policies indeed discover very different gaits and locomotion skills from each other and do not simply exploit some  source of randomness in the environment, even in high dimensional and action domains like the dog. Please let us know if these references address your concerns or is there any other experiment you suggest we should do.
>
> Thank you for pointing out the confusion between the single and multi agent problem formulations. While the problem of discovering a set of quality diverse policies has some multi-agent characteristics, it is a specific and somewhat redundant instance of the general multi-agent setting. Firstly, the discovered policies act in the environment independently of the other policies; their actions do not influence the actions of other policies. Markov games, on the other hand, allow both the transition dynamics and the reward signal to be a function of all the actions of all the agents in the environment. Thus, if we chose to model our problem as a Markov game it would be a very redundant one. Secondly, the 3-player game in section 3.3 does not involve a game between “equivalent players”, ie, it is not a game between policies, but a game between a policy, a cost player, and a Lagrange multiplier. While there are some game theoretic properties that characterize this game, they are in a sense simpler than the more general multi agent problem. For these reasons, we chose to extend the single agent MDP formulation instead of starting from a Markov game and limiting it to our scope.

---

> > ### Comment · Reviewer_9fff · 2022-11-25
> > **Response to Authors**
> >
> > Thank you for your response.
> >
> > >Thus, if we chose to model our problem as a Markov game it would be a very redundant one
> >
> > It's fine to have a very-redundant Markov game.  For clarity it would be good to note that it is *almost* a single-agent setting with just one change that makes it multi-agent.
> >
> > >  the 3-player game in section 3.3 does not involve a game between “equivalent players”, ie, it is not a game between policies, but a game between a policy, a cost player, and a Lagrange multiplier
> >
> > This is fine too.  There are often wildly asymmetric games.  But it is critical to point out that it is a game and not a fully single-agent setting!
> >
> > >For these reasons, we chose to extend the single agent MDP formulation instead of starting from a Markov game and limiting it to our scope
> >
> > Unfortunately, the theory and behavior of single and multi-agent systems are very very different.  For instance algorithms that converge in single agent settings often cycle forever or diverge in multi agent settings.  This the question of whether the tasks the RL agent is being asked to learn is, in fact, stationary is of critical importance, and if that task isn't stationary calling it an MDP risks forgetting important details.  Moreover, saying that the resulting object is an MDP, when it is actually a very degenerate Markov game that is almost an MDP is incorrect, and should be fixed just to ensure the claims in the paper are accurate.
> >
> > More concerning, is that since the problem is said to be a single-agent problem, even researchers who know about the intricacies of multi-agent learning would think that the final result avoids all of those intricacies.  That is they would think it would imply that certain algorithms converge, that there is a unique solution, and several other formal properties that come from having a single agent setting.  Thus it is very important to keep in mind that it is a game and not a single agent setting!

---

> > > ### Author Response · Authors · 2022-11-26
> > > **Response**
> > >
> > > Thank you for your response!
> > >
> > > This is indeed a delicate manner and we would like to further clarify our setup. Our work extends a recent line of work on MDPs with a general utility function, which are studied for convex utilities in [1,5,6,7]. In a standard MDP there is a reward signal and the objective is to maximize $d_\pi \cdot r$ (where r is the reward vector, see Section 2 for more details). However, in a convex MDP problem, the goal is to minimize a convex function of the state occupancy $f(d_\pi)$.
> > >
> > > Convex MDPs are indeed different objects from both standard MDP and Markov games. In fact, there is no reduction from a convex MDP to a standard MDP (see, for example, Lemma 1 in [1]). The environment in a convex MDPs is exactly the same as in a standard MDPs (single agent, Markov environment), and concretely, the distribution of the agent’s next state is specified by the current state and the action that it takes ($P(s'|s,a)$). That said, it is also possible to extend the convex utility setting to Markov games (see for example [8]) or to use multi agent algorithm to solve convex MDPs [5].
> > >
> > > Since these are different objects, as the reviewer pointed out, the algorithms for solving convex MDPs are different from those that are used to solve standard MDPs. Most of the algorithms for solving convex MDPs involve applying Fenchel duality to the objective, transforming it to a saddle point problem, and solving it as a game (for example, the 3-player game in section 3.3 ). We emphasise that this is not changing the problem definition (nor making it a multi agent problem). However, since a different class of algorithms is used to solve convex MDPs, the analysis of these algorithms and their convergence guarantees are indeed different. We also note that this is a common approach in Machine Learning, for example, many regularised objectives have an equivalent robust (saddle point) formulation [2,3,4].
> > >
> > > For these reasons, it is important to clarify the convergence guarantees for solving convex MDPs as saddle point problems. When both the maximizing and the minimizing players are online convex optimization algorithms (for example, the Follow the Leader cost player we use in the paper combined with an RL algorithm), then, the $\textbf{average}$ of the state occupancies (over the iterates of the algorithm) converges to the optimal solution (see Theorem 1, Lemma 2 and Lemma 3 in [1]). However, as the reviewer suggested, there are no guarantees on the last iterate convergence (that the current policy converges), and indeed, the iterates of the parameters may oscillate or diverge, but still converge on average!
> > >
> > > Thus, even a single agent, convex MDP setting suffers from all the issues that the reviewer mentioned. In our paper, we extended this setup to involve a set of policies. That said, the environment for each agent is still a single agent Markov environment and therefore does not introduce new challenges. Concretely, for an objective that is jointly convex in the state occupancies of a set of policies, with a single agent environment (the next state of each agent depends only on the current state of the agent and the action of the agent) all the policies will converge on average, similar to the single policy convex MDP setting.
> > >
> > > This is a great discussion and we agree with the reviewer that clarifying these details is important. We will make sure to add this discussion to the main paper.
> > >
> > > [1] Zahavy, T., O'Donoghue, B., Desjardins, G. and Singh, S., 2021. Reward is enough for convex MDPs. Advances in Neural Information Processing Systems, 34, pp.25746-25759.
> > >
> > > [2] Xu, H. and Mannor, S., 2012. Robustness and generalization. Machine learning, 86(3), pp.391-423.
> > >
> > > [3] Xu, H., Caramanis, C. and Mannor, S., 2008. Robust regression and lasso. Advances in neural information processing systems, 21.
> > >
> > > [4] J. D. Abernethy and J.-K. Wang. On frank-wolfe and equilibrium computation. In I. Guyon, U. V.
> > > Luxburg, S. Bengio, H. Wallach, R. Fergus, S. Vishwanathan, and R. Garnett, editors, Advances
> > > in Neural Information Processing Systems, volume 30. Curran Associates, Inc., 2017. URL https:
> > > //proceedings.neurips.cc/paper/2017/file/7371364b3d72ac9a3ed8638e6f0be2c9-Paper.pdf.
> > > [5] Geist, M., Pérolat, J., Laurière, M., Elie, R., Perrin, S., Bachem, O., Munos, R. and Pietquin, O., 2021. Concave utility reinforcement learning: the mean-field game viewpoint. arXiv preprint arXiv:2106.03787.
> > >
> > > [6] J. Zhang, A. Koppel, A. S. Bedi, C. Szepesvari, and M. Wang. Variational policy gradient method for reinforcement learning with general utilities. arXiv preprint arXiv:2007.02151, 2020.
> > >
> > > [7] E. Hazan, S. Kakade, K. Singh, and A. Van Soest. Provably efficient maximum entropy
> > > exploration. In International Conference on Machine Learning, pages 2681–2691. PMLR,
> > > 2019.
> > >
> > > [8] Ying, D., Ding, Y., Koppel, A. and Lavaei, J., Scalable Multi-Agent Reinforcement Learning with General Utilities.

---

### Official Review · Reviewer_PkBm · 2022-11-04

**Confidence:** 3
**Correctness:** 3
**Technical Novelty And Significance:** 3
**Empirical Novelty And Significance:** 2
**Recommendation:** 6

**Clarity, Quality, Novelty And Reproducibility:**

The paper is generally written appropriately with sections organized logically. The inner workings of the Constrained MDP optimizer are described without much technical detail in the main paper, but information needed for implementation (and sample code) is in the Appendix. In addition to the sample code, hyperparameters for all methods, including the proposed DOMiNO are given in the appendix. Some technical details are not described in the paper, including the implementation details of the policy gradient optimizer used for the policy, but a link to source code via the RLAX library is given, which appears to be sufficient for reproducibility.

The main components of the paper (1) two diversity objectives based on expected features, and (2) a framework for optimizing diversity while satisfying an expected reward constraint, appear to be novel, but it is possible I am not familiar with all necessary related works. In terms of quality, several weaknesses outweigh the paper’s strengths, including but not limited to the lack of baselines (see Weakness (4)). Comparisons to other diversity methods not proposed by the authors are lacking, and would significantly improve the quality of the paper if present.


**Strength And Weaknesses:**

Strengths Of The Paper:

1)

The diversity objective inspired by the Van Der Waals force is appealing because it encourages the set of policies to be diverse enough (as specified by a new hyperparameter referred to as the Van Der Waals contact distance), but not too diverse as to hinder reward maximization.

2)

Provided videos of the resulting policies show that diverse behaviors are indeed learned in all domains, which is particularly impressive in the DM-Control dog.walk domain.

3)

Ablations show that the method is not particularly sensitive to the size of the set of policies over which optimization is performed, which is encouraging since tuning this hyperparameter could otherwise require knowledge about the number of different gaits the agent can learn in advance.

Weaknesses Of The Paper:

1)

Results on K-shot adaptation when baselines are tuned for performance (see Figure E10 in the appendix), appear to show that SMERL with $\alpha=1.0$ performs as well as DOMiNO in all environments, and occasionally performs better such as in “Walk+Torso length”. The authors note that SMERL has the additional hyperparameter of $c_d$, while DOMiNO does not since the method uses constrained optimization, which can be seen as adaptively turning $c_d$.

2)

K-shot adaptation is a misleading title since in practice, for a set of 50 policies, if 10 trajectories are collected for each policy, this results in the agent collecting 500 trajectories. In the spirit of k-shot evaluation, this experiment could be made stronger if results were reported as K varies. Additionally, the authors are encouraged to be clear about how many environment transitions are collected in all K-shot adaptation experiments in the paper for reproducibility.

3)

While the ablations presented are sound, and the proposed framework is novel, the empirical results in the paper are weak: the proposed method, DOMiNO, does not perform distinguishably better than its alternatives, especially the SMERL implementation. The additional work of using constrained optimization, including a “cost player us[ing] the Follow the Leader (FTL) algorithm” (Page 3) does not provide a benefit over the simpler SMERL, which uses standard RL. Evaluation on more tasks could provide evidence to counter this interpretation, and I encourage the authors to provide additional results and clarifications on the comparison to SMERL.

4)

The proposed diversity objectives are not compared to other diversity objectives in the literature, such as DIAYN (Eysenbach et al., 2019), and MaxEnt RL (Haarnoja et al., 2018), which makes it hard to judge the utility this paper provides over existing methods for diversity maximization. It could be the case that the proposed method better controls the tradeoff between maximizing rewards and maximizing diversity (which may hinder reward maximization) via the Van Der Waals force. However, the paper is missing such a comparison, and it should be added.

Suggestions For The Paper:

1)

On page 5, the authors write “one could consider other combinations of powers and coefficients” for the diversity objective defined in Equation 7. While not necessary, and not currently negatively affecting my review (see the listed weaknesses above), the paper could be made stronger if an ablation of different powers and coefficients was given in the Appendix.


**Summary Of The Paper:**

This paper investigates how to learn near optimal policies with diverse gaits by augmenting a simple reward function in the domain with a diversity objective, which can be implemented as either constrained optimization, or as a multi-objective reward. The contributions of the paper include a diversity objective that repurposes the reward function from Abbeel & Ng (2004) to repel policies based on their expected feature similarity. Additionally, a second diversity objective is presented inspired by the Van Der Waals force, which extends the repulsive force to have a repulsive and attractive term---this is appealing because once learned policies are diverse enough, the Van Der Waals force is zero, and policies can purely maximize rewards.

The authors propose a framework for optimizing these objectives using Constrained MDPs, and provide ablations for a corresponding multi-objective variation of the objectives.The resulting method is called DOMiNO, and solves a constrained optimization problem, where the aim is to find a near optimal set of policies (where near optimal is defined in terms of recovering a fraction $\alpha$ of the performance of the optimal policy), while maximizing the diversity objective.


**Summary Of The Review:**

Overall, this paper studies an important problem in reinforcement learning: finding policies with diverse gaits that effectively balance diversity with task performance. The proposed method has appealing qualities: higher diversity, insensitivity to hyperparameters, and good task performance once diversity is attained. However, four significant weaknesses outweigh the paper’s strengths and consequently I cannot recommend the paper for acceptance in its current form. I am willing to reconsider my evaluation in light of new information and encourage the authors to address my concerns or provide clarifications where necessary.

---

> ### Author Response · Authors · 2022-11-15
> **Official Response**
>
>  We would like to thank the reviewer for finding the VDW reward interesting, for examining our videos and finding them to be demonstrating the discovery of quality diverse policies in challenging domains, and for appreciating our ablation studies. Next, we would like to address the reviewer’s concerns:
>
> 1. Both SMERL and the Lagrange multipliers are valid solutions to solve a constrained RL problem. As such, one should expect that they would find similar solutions when applied to the same problem, and indeed, our empirical results confirm that both methods find equivalently good solutions that satisfy the constrained problem and perform similarly well in K-shot adaptation.
>
> That said, learning the Lagrange multiplier instead of having to tune $c_d$ is an important contribution. It removes the need to tune a hyper parameter that SMERL is sensitive to. In general, tuning in the Kshot adaptation is not simple, as it is almost impossible to tune the algorithm based on the performance in the “base task”. Tuning (picking $\alpha=1$) as the reviewer suggested is only possible when looking at the kshot adaptation results. This is risking what is called p-hacking or reusing the holdout set (choosing the method that works best on the test set) so it requires more analysis in order to say that picking the best hyper parameter for SMERL is equivalent to the Lagrange multipliers.
>
>
> 2. K-shot adaptation. To address the reviewers' concerns we performed the following experiment which can be found in Subsection E.3 in our revised paper (Figure E10). We report the performance of the best policy in the set (Y axis) for different amounts K of total trajectories (X axis). Since K is the total number of trajectories, it is not always perfectly divided by the number of policies. Thus, we randomly distribute the modulo between the policies. The results show that using more trajectories clearly helps, but it also suggests that it saturates early on and that it is possible to use a much smaller K and still outperform the “single extrinsic policy baseline” (with ~30 trajectories or 2-3 per policy since the set is of size 10). Lastly, the length of each trajectory is exactly 1000 steps.
>
> 3. Deciding if SMERL is simpler than DOMINO is hard to answer.  Firstly, both methods use standard RL in the sense that they do not require any change to the RL algorithm other than providing it with a reward signal.While the SMERL paper was the first to propose constrained optimization to balance between a quality (extrinsic) reward, and a diversity (DIAYN like) reward, constrained optimization in RL dates back to much earlier than the SMERL paper, and the “go to” approach is to use Lagrange multipliers. Regarding the simplicity of the DIAYN like reward used in SMERL compared to our proposed rewards that are implemented via a FTL player. DIAYN requires training a discriminator and often involves tuning of the temperature of the softmax and the learning rate of the discriminator. The FTL player might seem complicated from an analysis point of view, but in practice it is straightforward to implement. Thus, we believe that DOMiNO which does not have this extra hyper parameter has an advantage on this front.
>
> In addition, as the other reviewers have mentioned, our paper proposes a novel optimization framework and new algorithms for quality diversity in RL and we believe that even if it looks more complicated at first glance, the fact that it can be used efficiently is an interesting contribution even if it performs similarly to previously proposed methods.
>
> 4. While we did not have enough time to implement a version of DIAYN in our code base during the rebuttal phase, we promise to add such a baseline to the final version of the paper.
>
> 5. We are working on adding this ablation and will hopefully upload it before the end of the discussion period. It is currently running, and based on the current results and previous experiments on this, we do not expect the algorithm to be sensitive to these hyper parameters. We chose the specific values to make the formula of the reward signal as simple as possible.

---

> > ### Author Response · Authors · 2022-11-16
> > **Following up on the "Suggestions For The Paper"**
> >
> > We have added a new section at the end of Supplementary C with Figures C9 (a,b) studying the effects the powers in the VDW force have on extrinsic reward and diversity accordingly. As we suggested before these hyper parameters have very little affect in practice.

---

> > > ### Comment · Reviewer_PkBm · 2022-11-18
> > > **Response to authors**
> > >
> > > I thank the authors for addressing my concerns about K-shot adaptation, and the comparison to previous works, including SMERL, and DIAYN. Considering this new discussion, and clarification on the K-shot adaptation experiment, I have raised my rating to a 6.

---

### Decision · Program_Chairs · 2023-01-20

**Decision:**

Accept: poster

**Justification For Why Not Higher Score:**

No reviewer was championing this paper.

**Justification For Why Not Lower Score:**

All reviewers have a consistent score of 6 and believe this is a good paper to be accepted.

**Metareview: Summary, Strengths And Weaknesses:**

The paper introduces a framework to learn a diverse set of policies with a controlled amount of sub-optimality. The paper introduces two diversity objectives, and solves the resulting problem using tools for solving convex MDPs. It should be mentioned that resulting problems are not necessarily convex MDP, so the success of the idea is evaluated only empirically.

We have a consensus among all reviewers that this is a good paper overall. There is still room for improvement though. For example, a more clear algorithmic description and better empirical analyses are suggested by the reviewers. I encourage the authors to consider the reviewers' feedback in order to improve the work. I recommend acceptance of the paper.

**Note From Pc:**

if the above contains the word "oral" or "spotlight" please see: "oral" presentation means -> notable-top-5% and "spotlight" means -> notable-top-25%. As stated in our emails, we are disassociating presentation type from AC recommendations